# Astroglial *Hmgb1* regulates postnatal astrocyte morphogenesis and cerebrovascular maturation

Moises Freitas-Andrade [1], Cesar H. Comin[2], Peter Van Dyken[3], Julie Ouellette[1,3], Joanna Raman-Nair[1,3], Nicole Blakeley[1,3], Qing Yan Liu[4,5], Sonia Leclerc[4], Youlian Pan [6], Ziying Liu [6], Micaël Carrier [7], Karan Thakur[1], Alexandre Savard[1], Gareth M. Rurak[8], Marie-Ève Tremblay [7], Natalina Salmaso[8], Luciano da F. Costa[9], Gianfilippo Coppola[10] & Baptiste Lacoste [1,3,11] ✉

Astrocytes are intimately linked with brain blood vessels, an essential relationship for neuronal function. However, astroglial factors driving these physical and functional associations during postnatal brain development have yet to be identified. By characterizing structural and transcriptional changes in mouse cortical astrocytes during the first two postnatal weeks, we find that high-mobility group box 1 (*Hmgb1*), normally upregulated with injury and involved in adult cerebrovascular repair, is highly expressed in astrocytes at birth and then decreases rapidly. Astrocyte-selective ablation of *Hmgb1* at birth affects astrocyte morphology and endfoot placement, alters distribution of endfoot proteins connexin43 and aquaporin-4, induces transcriptional changes in astrocytes related to cytoskeleton remodeling, and profoundly disrupts endothelial ultrastructure. While lack of astroglial *Hmgb1* does not affect the blood-brain barrier or angiogenesis postnatally, it impairs neurovascular coupling and behavior in adult mice. These findings identify astroglial *Hmgb1* as an important player in postnatal gliovascular maturation.

Astrocytes in the mature brain are physically and functionally coupled to blood vessels, a multicellular ensemble referred to as the gliovascular unit, which regulates cerebral blood flow and the blood-brain barrier to support neurotransmission and maintain neuronal health[1,2]. In mice, the genesis of astrocytes is initiated in the late prenatal period, and postnatally these glial cells proliferate rapidly and undergo extensive morphological changes[3–6], coinciding with the expansion of cerebrovascular networks[5,7–9]. During postnatal brain development, perivascular astroglial processes called endfeet contact blood vessel walls and the communication between these structures can mutually influence growth and maturation[10–12]. However, exactly when, and how, the brain gliovascular unit is established is not known.

During the first two postnatal weeks of brain development, astroglial morphogenesis is highly active, with primary branches sprouting from the cell body and dividing into finer processes[5,7]. Astroglial processes continue to ramify until a highly complex spongiform-like morphology is achieved, which requires extensive

[1]Neuroscience Program, The Ottawa Hospital Research Institute, Ottawa, ON, Canada. [2]Federal University of São Carlos, Department of Computer Science, São Carlos, Brazil. [3]Cellular & Molecular Medicine, University of Ottawa, Ottawa, ON, Canada. [4]National Research Council of Canada, Human Health and Therapeutics, Ottawa, ON, Canada. [5]Department of Biochemistry Microbiology and Immunology, Faculty of Medicine, University of Ottawa, Ottawa, ON, Canada. [6]Digital Technologies, National Research Council of Canada, Ottawa, ON, Canada. [7]Division of Medical Sciences, University of Victoria, Victoria, BC, Canada. [8]Department of Neuroscience, Carleton University, Ottawa, ON, Canada. [9]University of São Paulo, São Carlos Institute of Physics, FCM-USP, São Paulo, Brazil. [10]Yale School of Medicine, Dept. of Pathology, New Haven, CT, USA. [11]University of Ottawa Brain and Mind Research Institute, Ottawa, ON, Canada. ✉e-mail: blacoste@uottawa.ca

cytoskeletal remodeling. However, very little is known about the time course and molecular machinery of astroglial morphogenesis[13].

While astroglial endfeet are separated from endothelial cells and pericytes by the basal lamina, extensive signaling occurs between astrocytes, pericytes and endothelial cells[14–18]. Moreover, a recent study reported that a large portion of the astrocyte proteome is dedicated to astroglial endfeet, highlighting the complex nature of this cellular compartment[19]. Yet, the fundamental principles guiding endfoot placement around brain blood vessels remain elusive[20].

Neuronal activity and metabolism are inextricably "coupled" via the gliovascular unit[1,21,22]. Abnormal gliovascular growth has been observed in neurodevelopmental disorders and is associated with long-term neurological consequences[23–28]. Moreover, astrocyte dysfunction is linked to most, if not all, neuropsychiatric conditions[29,30]. For example, coverage of blood vessels by astroglial endfeet was significantly reduced in subjects with major depressive disorders[28]. Therefore, understanding the fundamental mechanisms of astrocyte morphogenesis and gliovascular maturation is crucial to develop strategies that promote healthy brain development. Here, we characterize the early postnatal temporal profile of cellular and molecular changes at the gliovascular interface in the mouse cerebral cortex, and we identify a mechanism that regulates gliovascular development. We show that a key molecular factor produced by astrocytes, namely HMGB1, regulates astrocyte morphogenesis via regulation of the cytoskeleton, controlling endfoot placement and endfoot protein distribution during postnatal brain development, with long-term implications in neurovascular coupling and behavior.

## Results

### Mouse brain astrocytes initiate vascular endfoot coverage during the first postnatal week

To thoroughly assess the temporal profile of gliovascular development and set the stage for subsequent molecular investigations, we employed transgenic mice producing the enhanced green fluorescent protein (EGFP) under the control of pan-astroglial Aldehyde Dehydrogenase 1 Family Member L1[31] (Aldh1l1) promotor. Analysis of EGFP-labeled astrocytes and CD31-labeled endothelial cells (ECs) showed that cortical astrocytes and microvessels increase in complexity over the first two postnatal weeks with little change during the third postnatal week (Fig. 1a, Supplementary Fig. 1a). Astrocyte morphogenesis was highly active during the first 5 postnatal days (Supplementary Fig. 1b), followed by an active phase of angiogenesis up to P14 (Fig. 1a, Supplementary Fig. 1a)[6,8,32]. At P0, perivascular astrocytes exhibited simple main branches aligned along the scaffold of blood vessels (Fig. 1a), and astrocyte cell bodies were in close apposition with microvessels compared to other timepoints (Fig. 1d). By P5, astroglial primary processes ramified into a highly branched morphology (Fig. 1b, Supplementary Fig. 1b). With cortical expansion, astrocytes positioned their cell body farther away from blood vessels and then progressively sent extensions back towards these vessels. This was also illustrated by the significant reduction in the proportion of astrocytes directly contacting vessels at P5 (Fig. 1c). After P5, astrocyte cell bodies remained distant from the vessel wall (Fig. 1d), but EGFP immunoreactivity on vessels increased sharply between P5 and P14 (Fig. 1e), consistent with a period of endfoot placement. Increases in astroglial extensions' branching and length continued until P21 (Supplementary Fig. 1b), and the dominant vertical orientation alongside vessels seen at birth was progressively replaced by a seemingly random orientation by P21 (Supplementary Fig. 1c). We also observed a rise in astrocyte cell density during that period (Supplementary Fig. 1a), consistent with recent findings[6,33], with a peak in proliferation of astrocytes around P4 correlating strongly with EC proliferation (Supplementary Fig. 1d). In addition, we observed glial acidic fibrillary protein (GFAP)-labeled radial glial processes at P5 and P7 (Supplementary Fig. 1a), coinciding with emergence of GFAP-positive astrocytes from P7, and with the

differentiation of radial glia into astrocytes in the developing cortex[34]. To confirm these observations obtained by immunofluorescence, we utilized transmission electron microscopy (TEM) to investigate postnatal development of the gliovascular interface at the ultrastructural level (Fig. 1f, Supplementary Fig. 2). TEM showed that microvessels were not yet contacted by perivascular endfeet at P0. By P7, however, large immature endfeet partially covered microvessels (Fig. 1f, h). Vascular coverage by endfeet observed at P14 via EGFP (Fig. 1a) translated into 100% coverage on TEM micrographs at the same age (Fig. 1f, h). Between P7 and P21, the number of astroglial endfeet increased, while the total endfoot area decreased (Fig. 1h), illustrating a refinement of these developing structures. Altogether this shows that perivascular endfoot placement establishes around P7 and is complete at P14, which is in line with observations made by Gilbert et al.[12].

### High mobility group box 1 (Hmgb1) is highly expressed in astrocytes at birth

Given the morphological changes observed concomitantly in astrocytes and microvessels between P0 and P14, we next sought to identify temporal changes in gene expression associated with gliovascular growth during that period. We first interrogated our published TRAP-Seq database[33], and then confirmed our observations using in situ multiplex RNA sequencing in astrocyte-enriched areas of interest (AOIs; Fig. 2a–c and Supplementary Fig. 3a–g). Consistent with major changes in astroglial morphology between P0 and P14, more genes appeared differentially expressed in astrocyte-enriched AOIs at P5 (Supplementary Fig. 4). When focusing on genes known as involved in cerebrovascular remodeling, we consistently found that Hmgb1 was highly expressed by astrocytes at birth (Fig. 2a, b), even though past work has focused on the role of this gene in the adult brain, where it is upregulated post-injury[35]. Expression of Hmgb1 in astrocytes was markedly higher than canonical pro-angiogenic genes, such as Vegfa or Wnt7a (Fig. 2a, b). Both Hmgb1 gene expression and HMGB1 protein levels decreased between P0 and P14 alongside maturation of the gliovascular unit (Fig. 2b, Supplementary Fig. 5a, b). HMGB1 protein was also produced by neurons and pericytes, but microglia and endothelial cells failed to display detectable HMGB1 nuclear immunoreactivity (Fig. 2e–g, Supplementary Fig. 5c, d). Yet, endothelial cells at P14 expressed Hmgb1 gene (mRNA), even at higher levels than astrocytes (Supplementary Fig. 9g), consistent with a recent adult mouse database[36,37] and suggesting important post-transcriptional regulation for this gene. Given the high expression of Hmgb1 in astrocytes at birth and the role of HMGB1 in adult neurovascular repair, we hypothesized that HMGB1 regulates astrocyte-dependent gliovascular maturation during this critical period. To test this, we generated Hmgb1^ΔAstro mice (Aldh1l1-CreER^T2+/−;Hmgb1^flox/flox) for conditional, astrocyte-specific ablation of Hmgb1 starting at P0 (Fig. 2h, and Supplementary Fig. 5e, f) and we confirmed selective HMGB1 removal from astrocytes as early as P7 (Fig. 2i). At P14, loss of HMGB1 in Hmgb1^ΔAstro astrocytes did not affect HMGB1 production by either neurons or pericytes (Fig. 3a, b and Supplementary Fig. 5d).

### Postnatal ablation of astroglial Hmgb1 affects endfoot maturation around blood vessels

Vessel density and branching appeared unaffected in Hmgb1^ΔAstro mice compared to control littermates at P14 (Supplementary Fig. 6a), suggesting that astroglial HMGB1 is not required for postnatal angiogenesis. While astrocyte cell density and surface ratio were unaltered in the cerebral cortex of Hmgb1^ΔAstro mice (Supplementary Fig. 6b), we found that distribution of gap junction protein connexin43 (Cx43), expressed by astroglial processes and endfeet[38,39], was significantly altered throughout the cerebral cortex of Hmgb1^ΔAstro mice (Fig. 3c). Disrupted distribution of Cx43 was particularly evident on larger microvessels (Fig. 3c, d). We also measured a ~3.5-fold increase in the number of astrocytes exhibiting high density "patches" of Cx43

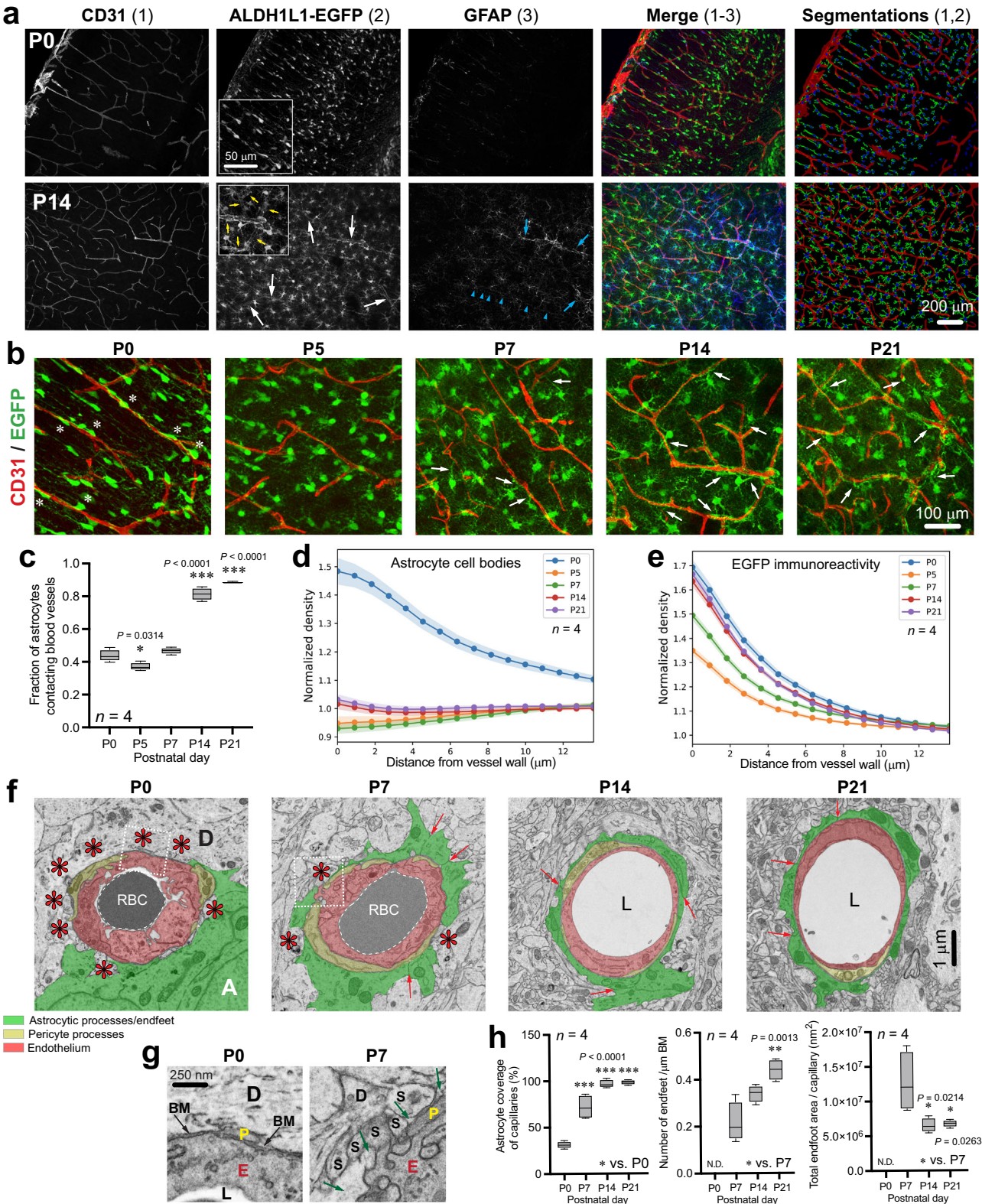

puncta in and around their cell body compared to control littermates (Fig. 3c, e). The number of clearly identifiable endfeet delineated by Cx43 on blood vessels was lower in *Hmgb1^{ΔAstro}* mice at P14 (Fig. 3c, e), while both Cx43 protein levels and total Cx43 immunoreactivity per vessel were unchanged (Fig. 3f, Supplementary Fig. 6c, d). Aquaporin-4 (AQP4), another astroglial endfoot marker, also displayed altered distribution with significant decrease in vascular coverage in *Hmgb1^{ΔAstro}* mice at P14 compared to age-matched controls (Fig. 3g).

We also found that AQP4 protein levels were unaffected in *Hmgb1^{ΔAstro}* mice (Fig. 3h), confirming that these alterations were in AQP4 distribution but not production. We then tested whether these changes in endfoot-enriched proteins were associated with altered endfoot placement around microvessels. Using immunofluorescence, we measured a significantly altered distribution of ALDH1L1 immunoreactivity onto CD31^+ microvessels (Fig. 4a). In *Hmgb1^{ΔAstro}* mice at P14, a higher number of vessels displayed reduced astroglial coverage, whereas a

**Fig. 1 | Gliovascular maturation during postnatal development of the cerebral cortex. a** Fluorescence micrographs of somatosensory cortex from *Aldh1l1-eGFP* male mice at P0 and P14. Endothelial marker CD31 (red); astrocyte markers EGFP (green) and GFAP (blue). Arrows point to astroglial endfeet around microvessels. Blue arrowheads indicate radial glia. Segmentations represent reconstructed astrocytes and vessels quantified to evaluate vessel density and branching, as well as astrocyte density and fraction contacting vessels. **b** Fluorescence micrographs of somatosensory cortex from *Aldh1l1-eGFP* mice at P0, P5, P7, P14 and P21. CD31 (red); EGFP (green). White asterisks indicate perivascular astrocyte cell bodies closely associated with microvessels at P0. White arrows indicate astroglial endfeet extending and contacting microvessels from P7. **c** Segmentations (see in **a**) were quantified to evaluate the fraction of astrocytes contacting vessels. Data are whisker boxes (min to max, center line indicating median). *$p < 0.05$, ***$p < 0.001$ (One-way ANOVA and Tukey's post-hoc test). **d, e** Quantifications of astrocyte cell body density (**d**) or EGFP immunoreactivity (**e**) as a function of the distance (µm)

from blood vessel wall between P0 and P21. Data are mean ± SEM. **f** Electron micrographs show temporal developmental profile of astrocyte coverage (pseudocoloured in green) around microvessels (pseudocoloured in red). Pericytes can also be observed (pseudocoloured in yellow). Red asterisks indicate areas lacking astroglial coverage. Red arrows point at inter-endfoot walls. A astrocyte, D neuronal dendrite, L lumen, RBC red blood cells. **g** Higher magnifications corresponding to dashed boxes in (**f**) illustrating fine features of astrocytic endfeet at zones lacking endfoot coverage. BM basement membrane, D neuronal dendrite, E endothelium, L lumen, P pericyte, S dendritic spine. Black arrows, BM. Green arrows point at astrocyte endfoot cytoplasm. **h** Analysis at the nanoscale level of astrocyte maturation and endfoot coverage during postnatal brain development. Data are whisker boxes (min to max, center line indicating median) in (**g**). *$p < 0.05$, **$p < 0.01$, ***$p < 0.001$ (One-way ANOVA and Tukey's post-hoc test). All displayed microscopy images are representative of experiments repeated in 4 mice per group, with similar results. Source data are provided as a Source Data file.

lower number of vessels displayed elevated (i.e. normal) coverage compared to controls. At the ultrastructural level (TEM), endfoot morphology appeared disorganized in *Hmgb1^ΔΔAstro^* mice and exhibited a ~1.5-fold increase in the number of smaller, fragmented endfeet around microvessels, with some perivascular areas lacking astroglial coverage (Fig. 4b, Supplementary Fig. 7a, b). Furthermore, endothelial morphology appeared dramatically altered in *Hmgb1^ΔΔAstro^* mice at P14, with increased number (2–3-fold) of vascular profiles displaying macropinocytic extensions and vacuoles (Fig. 4b, Supplementary Fig. 7b, c). We also observed several instances where endothelial protrusions extended into astroglial endfeet, but only in *Hmgb1^ΔΔAstro^* mice (Supplementary Fig. 7d). Given the morphological alterations measured in endothelial cells from *Hmgb1^ΔΔAstro^* mice at P14, we sought to determine whether blood-brain barrier (BBB) integrity was compromised at this age in mutant animals. Injection of small tracer AlexaFluor555-conjugated Cadaverine (~1 kDa) did not reveal any leakage in the cerebral cortex of *Hmgb1^ΔΔAstro^* mice compared to controls (Fig. 4c). Moreover, levels of tight and adherens junction proteins appeared similar between mutant and control mice (Fig. 4d, Supplementary Fig. 7e). In addition, despite disturbed endothelial morphology, tight junction ultrastructure and the total number of caveolae-type vesicles appeared normal in *Hmgb1^ΔΔAstro^* mice (Supplementary Fig. 10a), altogether suggesting an intact BBB despite endfoot misplacement.

### Postnatal ablation of astroglial *Hmgb1* affects astrocyte morphogenesis

Given the astroglial endfoot phenotype observed in *Hmgb1^ΔΔAstro^* mice, we assessed whether loss of HMGB1 affects overall astroglial morphology in vivo and in vitro. First, a systematic in vivo morphological analysis was performed on ALDH1L1-immunostained brain sections from mutant and control littermates at P14 (Fig. 5a, Supplementary Fig. 8). Astrocytes from *Hmgb1^ΔΔAstro^* mice displayed a significant decrease in several morphological metrics, including reduced number and total length of main branches (Fig. 5a). Second, to determine whether aberrant astrocyte morphology is a cell-autonomous phenotype, primary astrocyte cultures were prepared from *Hmgb1^ΔΔAstro^* and control littermates and immunostained for GFAP and HMGB1 (Fig. 5b, c). Similar to what was observed in vivo, primary astrocytes isolated from *Hmgb1^ΔΔAstro^* mice exhibited significant changes in morphology, with less extensions and a more flattened cell body compared to control astrocytes (Fig. 5d, e). Considering the roles played by ECs and astrocytes (including Cx43) in the maturation of the cerebral cortex and neuronal networks[11,24,40], immunofluorescent analysis of three neuronal markers (NeuN, CTIP2 and POU3F2) was performed to test whether lack of astroglial *Hmgb1* affects cortical growth. No difference in neuronal layering across cortical depth was observed between the *Hmgb1^ΔΔAstro^* and control mice at P14 (Supplementary Fig. 9a), altogether demonstrating that astroglial *Hmgb1* is required for proper

postnatal maturation of astrocytes and that this is independent of cortical lamination.

### Astroglial *Hmgb1* regulates the expression of genes related to cytoskeletal remodeling

Next, to unmask the molecular underpinnings of altered gliovascular structure at P14 in *Hmgb1^ΔΔAstro^* mice, we isolated primary cortical astrocytes and ECs from *Hmgb1^ΔΔAstro^* and control littermates and performed deep RNA sequencing on each cell type from individual mice (Fig. 6a, b and Supplementary Fig. 9b, d). In contrast to the markedly disturbed ultrastructure of ECs in *Hmgb1^ΔΔAstro^* mice at P14, transcriptomic analysis showed little-to-no change in endothelial gene expression (Supplementary Fig. 9c). Compared to ECs, astrocytes from *Hmgb1^ΔΔAstro^* mice displayed a larger number of differentially-expressed genes (DEGs) (Fig. 6c, d). In particular, four astroglial DEGs downregulated in *Hmgb1^ΔΔAstro^* astrocytes have been associated with cytoskeletal remodeling, namely *Slit1*, involved in neuronal morphogenesis[41,42], *Camk1g* encoding Calcium/calmodulin-dependent protein kinase Iγ (CaMKIγ) which regulates activity-dependent dendritic growth via cytoskeletal remodeling[43,44], *Arhgef28* encoding Rho guanine nucleotide exchange factor (RGNEF) involved in microtubule network remodeling[45,46], as well as *Ptger4* encoding prostaglandin E receptor 4 (EP4) which controls actin cytoskeleton remodeling[47] (Fig. 6e, Supplementary Fig. 9d). Of note, activation of EP4 was shown to inhibit actin ring formation in osteoclasts[47]. Consistent with these observations, and possibly linked to *Ptger4* downregulation, we consistently observed significantly more *Hmgb1^ΔΔAstro^* astrocytes presenting a somatic actin ring compared to control astrocytes when stained with F-actin-binding phalloidin (Fig. 6f, Supplementary Fig. 9e, f). Fewer transcripts were significantly upregulated (FRD < 0.05; FC > 2) in *Hmgb1^ΔΔAstro^* astrocytes, including *Teddm2*, encoding the epididymal protein Me9, and *Lurap1l*, predicted to function as an adaptor involved in the regulation of cell shape[48]. Collectively, these data suggest that HMGB1 regulates cytoskeletal arrangement in postnatal astrocytes.

### Loss of astroglial *Hmgb1* postnatally affects cerebrovascular function and behavior into adulthood

Considering the morphological and transcriptional alterations measured in postnatal astrocytes from *Hmgb1^ΔΔAstro^* mice, we sought to determine consequences of loss of astroglial HMGB1 on gliovascular unit function in adult *Hmgb1^ΔΔAstro^* animals. We assessed neurovascular coupling in the primary somatosensory (S1) cortex of young adult (P50) *Hmgb1^ΔΔAstro^* mice and control littermates. Upon whisker stimulation in *Hmgb1^ΔΔAstro^* mice, we measured reduced cerebral blood flow (CBF) responses (% increase), as well as reduced CBF response amplitude during stimulation (Fig. 7a). Other hemodynamic parameters were unaffected by the conditional mutation (Supplementary Fig. 10c). The reduction in CBF response exhibited by *Hmgb1^ΔΔAstro^* mice appeared independent of neuronal activation, as no change in cFos induction

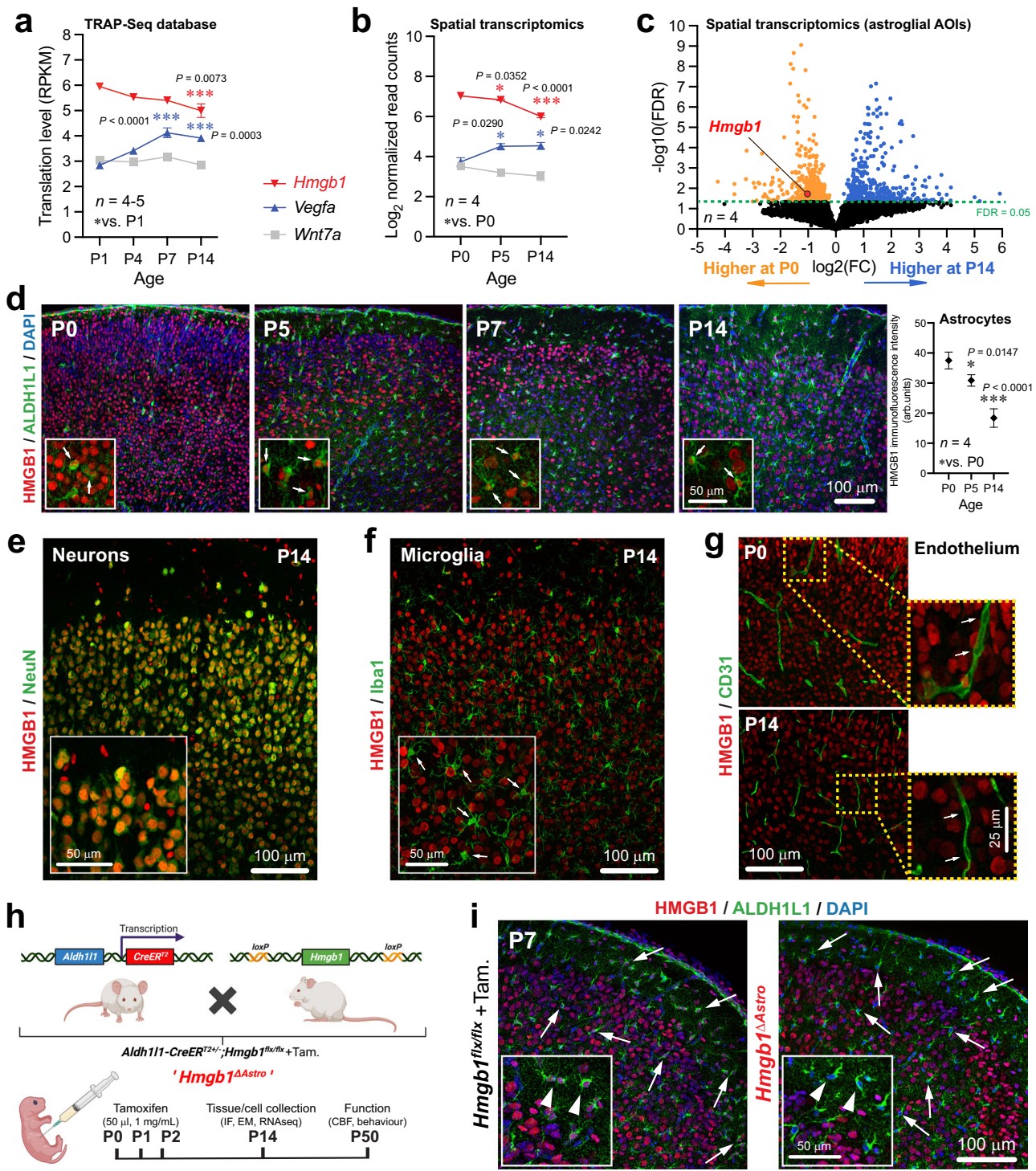

was observed in S1 cortex of whisker-stimulated *Hmgb1[ΔAstro]* animals (Fig. 7a, Supplementary Fig. 10c). We also tested whether altered CBF in adult mutants correlated with aberrant distribution of cortical Cx43. Consistent with our findings at P14, the Cx43 phenotype persisted in adult *Hmgb1[ΔAstro]* mice compared to age-matched controls, yet the size of abnormal Cx43 puncta patches appeared higher at P50 compared to P14 in *Hmgb1[ΔAstro]* mice (Fig. 7b). However, protein levels of Cx43 were significantly lower in mutant mice compared to littermate controls (Supplementary Fig. 10d). In addition, protein levels of AQP4 appeared slightly higher in P50 mutant mice compared to age-matched controls (Supplementary Fig. 10d). Of note, no astrogliosis, was found in either control or mutant mice at P50 as assessed by low GFAP

immunoreactivity (Supplementary Fig. 10b). Finally, we hypothesized that all the alterations reported above in *Hmgb1[ΔAstro]* mice may lead to behavioral changes in adult mice. *Hmgb1[ΔAstro]* mice displayed significantly reduced marble-burying behavior compared to control littermates (Fig. 7c), an indicator of stereotypic and/or repetitive behavior. While *Hmgb1[ΔAstro]* and control mice showed comparable behaviors in the open field and novel object recognition tasks (Fig. 7d, Supplementary Fig. 10e, f), mutant mice travelled more distance during the 5-minute novel object test trial (Fig. 7d). In the elevated plus maze test, more entries in closed arms were measured for *Hmgb1[ΔAstro]* mice compared to their control littermates (Fig. 7e, Supplementary Fig. 10g), suggesting an anxiety-like phenotype in mutant animals.

**Fig. 2 | HMGB1 is highly expressed in astrocytes around birth in the mouse cerebral cortex. a** Expression (reads per kilo base per million mapped reads, RPKM) of *Hmgb1* relative to known pro-angiogenic astroglial genes at P1, P4, P7 and P14 from Star Database (https://stardb.cu-bic.ca/). Data are mean ± SEM. *$p < 0.05$, **$p < 0.01$, ***$p < 0.001$ (One-way ANOVA and Tukey's post-hoc test). Comparisons were made with respect to P1. **b** In situ expression (normalized read counts) of *Hmgb1* relative to known pro-angiogenic genes at P0, P5 and P14 in astrocyte-enriched (ALDH1L1⁺CD31⁻) areas of interest (AOIs). Data are mean ± SEM. *$p < 0.05$, **$p < 0.01$, ***$p < 0.001$ (One-way ANOVA and Tukey's post-hoc test). Comparisons were made with respect to P0. **c** Volcano plot (False discovery rate, FDR, versus Log2 fold change, FC) to visualize astroglial genes with differential expression between P0 and P14 in situ. **d** Left, Fluorescence micrographs of immunostained somatosensory cortex sections at specific time-points during postnatal development. Insets displayed higher magnification. Arrows indicate HMGB1 immunoreactivity (red) in ALDH1L1⁺ astrocytes (green). *Right*, Quantification of astroglial HMGB1 immunoreactivity at P0, P5 and P14. Data are mean ± SEM. *$p < 0.05$, ***$p < 0.001$ (One-way ANOVA and Tukey's post-hoc test). **e** Fluorescence micrograph showing HMGB1 (red) in NeuN⁺ neurons (green) at P14. Inset displayed higher magnification. **f** Fluorescence micrograph showing lack of HMGB1 (red) in Iba1⁺ microglia (green) from P14 mice. Inset displayed higher magnification, and arrows point at microglial cells. **g** Fluorescence micrographs of somatosensory cortical sections immunostained with CD31 (green) and HMGB1 (red) at P0 and P14. Insets displayed higher magnifications. **h** Diagram of mating strategy and tamoxifen injections for selective *Hmgb1* ablation from astrocytes, with experimental end points. **i** Fluorescence micrographs showing immunostaining for ALDH1L1⁺ astrocytes (green) and HMGB1 (red) in control and *Aldh1l1-CreER^{T2};Hmgb1^{flx/flx}* (or *Hmgb1^{ΔΔAstro}*) mice. Insets displayed higher magnifications. Arrows and arrowheads point to astrocytes with (left) or without (right) HMGB1. All displayed microscopy images are representative of experiments repeated in at least 4 mice per group, with similar results. Source data are provided as a Source Data file.

## Discussion

In this study, we analyzed gliovascular development, focusing on the time course of astrocyte maturation and endfoot placement around cortical microvessels, as well as on the cellular and molecular maturation of astrocytes with respect to cerebrovascular growth. We found that the first seven days after birth are critical for the maturation of astrocytes and for recruitment of their endfeet at the microvascular wall in the mouse cerebral cortex. We reveal that perivascular endfoot coverage begins to establish around P7 in the cerebral cortex, and that the complexity of astrocyte morphology and transcriptome evolves alongside postnatal growth of cortical vessels. At P5, marked transcriptional changes are apparent in astrocytes compared to other time-points during postnatal brain growth. Genes associated with cytoskeletal remodeling, such as *Mapt*[49], *Rasgrf2*[50] and *Gap43* linked to astrocyte arborization and elongation[51], highlight the dynamic nature of astrocyte morphogenesis early after birth. Prior to our work, several studies had reported elements of postnatal astrocyte maturation[4,6,52,53], but the precise time course underlying gliovascular maturation in the brain was unknown. Gilbert et al.[12] reported a mechanism by which the astrocyte-specific membrane protein MLC1 is critical for astrocyte orientation and polarity towards developing vessels[12], which could be at play during postnatal gliovascular growth. Another recent study focusing on the retinal vasculature reported that constitutive lack of *Adenomatosis polyposis coli downregulated-1* (*Apcdd1*) in mice led to precocious maturation of astrocytes with increased *Aqp4* expression and more extensive perivascular endfoot coverage at P14[54]. Testing whether *Hmgb1* expression is modulated in *Apcdd1^{-/-}* or *Mlc1^{-/-}* mice could provide a valuable genetic framework to study gliovascular maturation.

Our investigation into molecular players associated with gliovascular unit formation led us to identify HMGB1 as a molecular factor in astrocytes to regulate both astroglial morphogenesis and gliovascular unit maturation. HMGB1 is an evolutionarily conserved non-histone protein that is widely present in the nucleus of eukaryotic cells. It plays important roles in stabilizing DNA and regulating transcription[55]. HMGB1 has multiple functions depending on its location in the nucleus or cytosol, and it can be released actively from stressed cells, or passively after necrotic cell death[56]. Under pathological conditions in the adult brain, HMGB1 is upregulated and secreted to promote reparative angiogenesis and induction of inflammatory pathways[35,57–59]. In our study, *Hmgb1* gene was highly expressed at birth (i.e., in a developing, healthy brain) and rapidly decreased by the end of the second postnatal week. We propose that increased production of HMGB1 under pathological conditions in the adult brain relates to the induction of a developmental genetic program promoting growth and repair. Similarly, others have shown that developmental processes are reactivated in pathological conditions in adults[60–62], however a role for astroglial HMGB1 in neurodevelopment has been overlooked. Few studies using mice constitutively lacking HMGB1 (*Hmgb1^{-/-}*), or *Hmgb1* knockdown in zebrafish, reported aberrant brain development, with altered neural progenitor proliferation and survival[63,64]. Here, we provide evidence revealing an important role for astroglial *Hmgb1* in gliovascular development.

Postnatally, *Hmgb1^{ΔΔAstro}* mice displayed profound morphological changes at the gliovascular unit (both in astrocytes and ECs) as revealed by in vivo and in vitro experiments. However, the BBB was not affected in these mutant mice, since neither small-molecule leakage nor tight junction protein difference were observed between *Hmgb1^{ΔΔAstro}* mice and their control littermates. Previous studies have shown that removal of astroglial endfeet does not perturb the BBB, consistent with our findings in *Hmgb1^{ΔΔAstro}* mice at P14. On the one hand, the gliovascular unit exhibits plasticity for instance upon brain injury, and astroglial endfeet replacement around blood vessels has been observed after laser ablation of endfeet[65] or following two-photon chemical apoptotic ablation of perivascular astrocytes[66]. However, whether single astrocytes are forming more vascular contacts, as a compensatory mechanism, or more astrocytes are sending processes to the same vascular portion *Hmgb1^{ΔΔAstro}* mice remains to be elucidated. We found that while *Hmgb1^{ΔΔAstro}* astrocytes displayed reduced main branches in vivo and in vitro, the number of smaller astrocytic endfeet appeared increased in vivo using TEM; This discrepancy may reflect the limitation in resolution of immunofluorescence (IF) detection by algorithms to fully delineate morphological characteristics of single astrocytes. In addition, it will therefore be important to verify if endfoot replacement can be observed in adult *Hmgb1^{ΔΔAstro}* mice. On the other hand, our observations imply that a contact-dependent regulation of the BBB by astroglial endfeet during postnatal development is unlikely. However, this does not rule out a regulation of the BBB by factors released from astrocytes, with implications for neurological disorders[67].

Several genes linked to cytoskeletal regulation were identified as significantly downregulated in *Hmgb1^{ΔΔAstro}* astrocytes, including *Slit1*, *Camk1g* and *Arhgef28*. In particular, *Ptger4*, encoding EP4 which regulates actin ring formation[47], was downregulated in *Hmgb1^{ΔΔAstro}* astrocytes. This observation correlated with a high proportion of *Hmgb1^{ΔΔAstro}* astrocytes exhibiting actin rings in cell culture compared to control cells. While very little is known about the function of HMGB1 in astrocytes, a recent in vitro study showed that overexpression of nuclear HMGB1 in astrocytes significantly increased the methylation of the SAM and SH3 domain-containing 1 (SASH1) gene, encoding a scaffolding protein associated with cytoskeletal regulation, and this was linked to reduced cell adhesion and increased migration[68].

One hallmark of astrocyte differentiation is the robust expression of Cx43, the most abundant connexin in the brain, involved in extensive coupling between astrocytes via gap junction[69]. This coupling allows for rapid communication between astroglial networks in response to neuronal demands[70–72]. In the cortex of *Hmgb1^{ΔΔAstro}* mice, despite the difference in Cx43 protein levels at P50, Cx43 protein

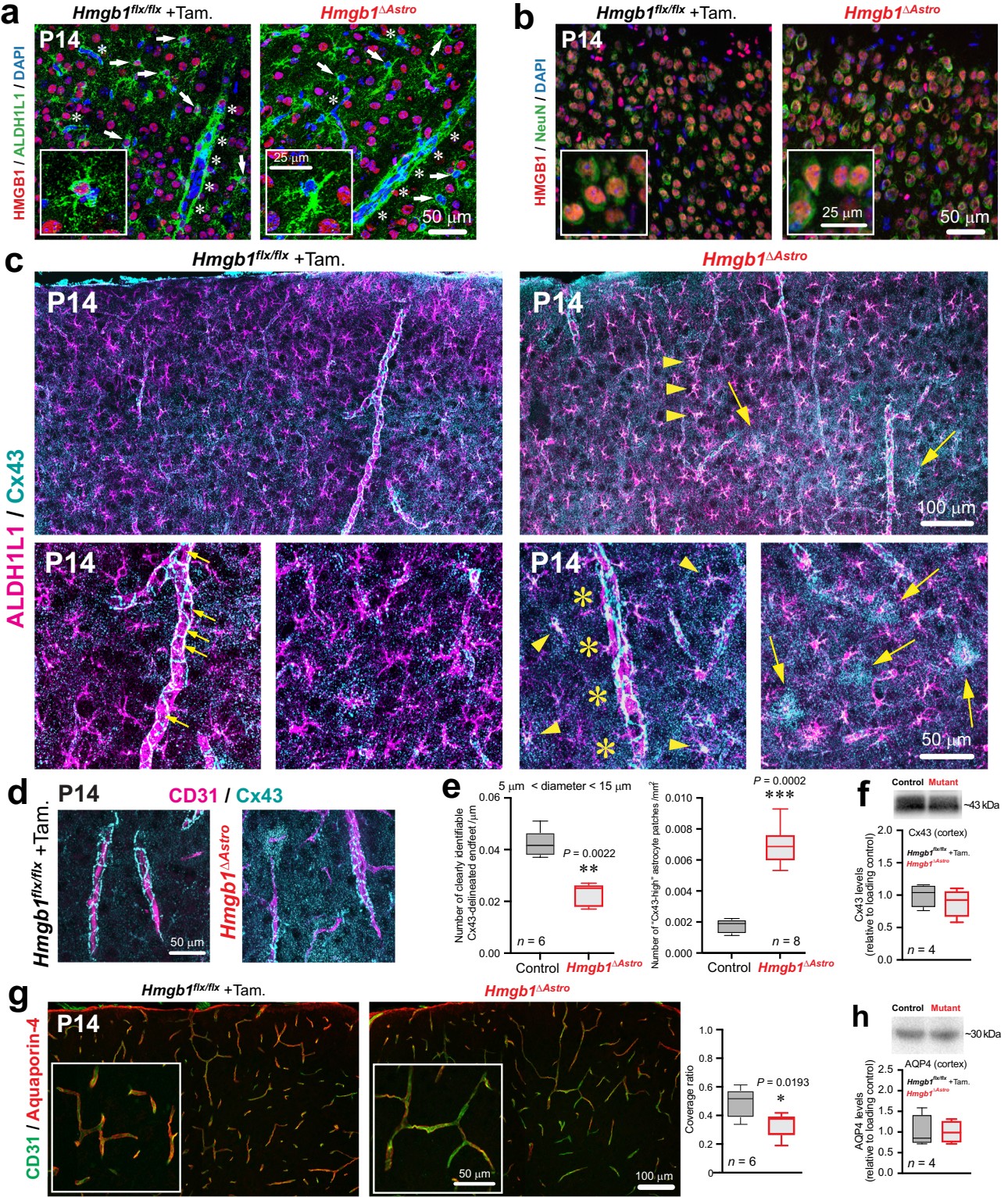

distribution appeared severely impaired at both P14 and P50. This might be explained by the aberrant cytoskeletal rearrangement in *Hmgb1^(ΔAstro)* astrocytes. It is possible that F-actin rings prevented the distribution of Cx43 beyond the cell body into astroglial extensions. Indeed, the C-terminal tail of Cx43 interacts with the actin cytoskeleton, a substrate for proper transport to the cell membrane[40,73]. Cx43 was also reported to control astrocyte morphogenesis via a channel-independent mechanism likely mediated by its adhesive properties[74]. Of note, HMGB1 was shown to promote differentiation of spinal ependymal cells into astrocytes[75]. Taken together, HMGB1 may function as a transcriptional regulator in developing astrocytes to regulate cytoskeletal reorganization and cellular morphogenesis.

The reduced morphological complexity of *Hmgb1^(ΔAstro)* astrocytes, observed both in vitro and in vivo, affected endfoot placement at the gliovascular interface, as measured at the TEM level and by endfoot markers Cx43 and AQP4. Interestingly, the spaces let by the *Hmgb1^(ΔAstro)* endfeet on blood vessels (TEM images) are small in comparison to the length of vessels with no AQP4 signal (immunofluorescent images) in

**Fig. 3 | Ablation of HMGB1 in newborn astrocytes affects astroglial endfoot protein distribution in the postnatal cerebral cortex. a** Fluorescence micrographs showing immunostaining for ALDH1L1[+] astrocytes (green) and HMGB1 (red) at P14. Asterisks delineate blood vessels. Insets are higher magnifications. Arrows point to astrocytes expressing (*left*) or lacking (*right*) HMGB1. **b** Fluorescence micrographs of neurons (NeuN, green) and HMGB1 (red). **c** Top left panel: Fluorescence micrograph showing normal Cx43 immunolabelling (cyan) in astrocytes (magenta) in control (*Hmgb1^flx/flx*) mice. Top right panel: Fluorescence micrograph showing disrupted Cx43 distribution in *Hmgb1^ΔΔAstro* mice. Arrowheads indicate increased Cx43 within astrocyte cell bodies. Arrows point to patches of dense Cx43 immunoreactivity. Bottom left panels: higher magnifications of Cx43 weblike patterns (small arrows) delineating endfeet along a vessel. Bottom right panels: higher magnifications of Cx43 disruption (yellow asterisks), and of Cx43-high patches (yellow arrows), on astrocytes. **d** Fluorescence micrographs showing Cx43 (cyan) on CD31[+] blood vessels (magenta) from control (left) or *Hmgb1^ΔΔAstro* (right) mice. **e** Average number of clearly-identifiable, Cx43-delineated endfeet (left) and Cx43

patches (right). Data are whisker boxes (min to max, center line indicating median). **p < 0.01, ***p < 0.001 (two-tailed Mann–Whitney's test). **f** Cx43 immunoblot of mouse cerebral cortex from control (n = 4) and *Hmgb1^ΔΔAstro* (n = 4) mice. Graph shows quantification of immunoblot signal normalized to the loading control and relative to control values. Data are whisker boxes (min to max, center line indicating median). **g** Left panels, Fluorescence micrographs showing aquaporin-4 immunoreactivity (red) on CD31[+] blood vessels (green) from control (left) or *Hmgb1^ΔΔAstro* (right) brain sections. Right panel, coverage ratio of aquaporin-4 on CD31[+] blood vessels. Data are whisker boxes (min to max, center line indicating median). *p < 0.05 (two-tailed Mann–Whitney's test). **h** Aquaporin-4 immunoblot in mouse cerebral cortex from control (n = 4) and *Hmgb1^ΔΔAstro* (n = 4) mice. Graph shows quantification of immunoblot signal normalized to the loading control and relative to control values. Data are whisker boxes (min to max, center line indicating median). All microscopy images displayed in this figure are representative of experiments repeated in at least 6 mice per group, with similar results. Source data are provided as a Source Data file.

*Hmgb1^ΔΔAstro* mice; However, we did not detect differences in AQP4 protein levels. Taken together, like the aberrant distribution observed for Cx43 in *Hmgb1^ΔΔAstro* astrocytes, AQP4 distribution at the endfeet (but not production) may have been affected by HMGB1 ablation. Of note, F-actin cytoskeleton plays a primary role for AQP4 plasma membrane localization and during cell adhesion[76]. While vessel branching and density were not affected by loss of astroglial HMGB1, suggesting normal angiogenesis, brain EC morphology was also severely impaired in P14 *Hmgb1^ΔΔAstro* mice. Brain ECs in P14 *Hmgb1^ΔΔAstro* mice were reminiscent of ECs from newly formed capillaries[77], and appeared morphologically closer to ECs we observed in wild-type brains at P0. Yet, little-to-no transcriptional change was measured in P14 *Hmgb1^ΔΔAstro* ECs, suggesting a post-transcriptional (contact-dependent or via released signals) regulation of EC maturation by astrocytes.

Changes observed at the gliovascular unit in *Hmgb1^ΔΔAstro* mice at P14 were followed by neurovascular coupling abnormalities in adults. This may be due to either aberrant Cx43 signaling in astrocytic endfeet[78,79] or abnormal astrocyte coverage affecting vascular smooth muscle cell contractility[12,80]. Moreover, adult *Hmgb1^ΔΔAstro* mice displayed behavioural changes, with reduced repetitive movements, increased locomotion in a cognitive task and more entries in closed arms of the elevated plus maze, overall indicative of increased anxiety. While we did not draw causal links, we propose that neurovascular deficits and altered distribution of Cx43, which persisted in adult *Hmgb1^ΔΔAstro* astrocytes, may contribute to the behavioural phenotype. Of note, Cx43 plays an important role in healthy synaptic transmission and cognition in mice[39]. The long-term implications associated with loss of astroglial HMGB1 on aging and resilience to brain injury remain to be established.

Cerebrovascular dysfunction, resulting from altered neurovascular maturation or neurovascular uncoupling, is a precipitating factor in neurological disorders[81,82]. Cognitive decline with reduced cerebral blood flow has been described as an early marker of vascular dementia[83] and Alzheimer's disease[84], both of which were linked to astrocyte dysfunction[29]. Cerebrovascular impairments in neurodevelopmental conditions such as autism spectrum disorders[24] and schizophrenia[85] have also been associated with astrocyte malfunction[86,87] and reduced cerebral blood flow in adults[26]. As aberrant astrocyte morphogenesis and gliovascular development may predispose the adult brain to cognitive decline and/or impair vascular responses to neuronal injury, elucidating the role of HMGB1 in gliovascular plasticity may lead to therapies for neuroprotection.

## Methods
### Animals
All animal procedures including euthanasia were approved by the University of Ottawa Animal Care Committee and were conducted in

accordance with the guidelines of the Canadian Council on Animal Care (Breeding Protocol: CMM-3317; Experimental Protocol: CMM-3956).

**Mouse husbandry.** All mice were bred in house and housed maximum five per cage with free access to food and water. Mice are on a 12/12 light cycle (7AM On / 7PM Off). Mice undergoing behavioral testing are on an inverted light cycle. Animal temperature for rodent rooms is 21 °C–23 °C, with humidity of 40–60%. To assess astrocyte coverage/density, *Aldh1l1-eGFP* (BAC) males (Jackson laboratory, Stock No. 026033; B6 background) were crossed with WT "Noncarrier" females (Jackson laboratory). Conditional knockout of *Hmgb1* was achieved by breeding *Aldh1L1-Cre/ERT2* BAC transgenic male animals[88] (Jackson laboratory, Stock No. 031008; C57BL/6N-congenic background) into a background of female mice carrying a loxP-flanked *Hmgb1* gene (*Hmgb1^flx/flx*)[89] (Jackson laboratory, Stock No: 031274; B6 background). Animals were backcrossed to generate *Hmgb1^flx/flx* offspring with or without *Aldh1L1-Cre/ERT2*. Offspring and breeding pairs were all confirmed by PCR. Littermate mice of both sexes were included in experiments at age postnatal day (P)0, P5, P7, P14, P21 or P50, as specified in the text and figure legends. For tissue extraction, pups 7-day-old and younger were decapitated using sharp scissors. For animals 14-day-old and older, anesthesia (Ketamine/xylazine, 100 mg/kg and 10 mg/kg respectively, i.p.) was used prior to euthanasia by cervical disclocation.

**Genotyping.** Heterozygous *Aldh1l1-eGFP* offspring were identified through PCR. Primers used were: 5′-GAACAGGCGAAAGCGTTAAG-3′ (23136, forward), 5′-GTAAACCTCCTGGCCAAACA-3′ (18707, reverse), 5′- CTAGGCCACAGAATTGAAAGATCT-3′ (oIMR7338, forward), and 5′-GTAGGTGGAAATTCTAGCATCATCC-3′ (oIMR7339, reverse) with PCR products of 550 bp (transgene) and 324 bp (internal positive control). For Cre genotyping, the following primers were used: 5′-GCAAGTT-GAATAACCGGAAATGGTT-3′ (forward) and 5′-AGGGTGTTA-TAAGCAATCCCC AGAA-3′ (reverse), with a 250-bp PCR product. For *Hmgb1^flx* genotyping, 5′-AAAGTTTGATGCGAACACG-3′ (forward) and 5′-TGATCTCAAGAGTAGGCACAGG-3′ (reverse), with a 400-bp mutant and 328-bp WT PCR product.

### Immunohistochemistry and immunofluorescence
**Immunofluorescence for two-dimensional (2D) imaging.** For postnatal development of astrocyte and vascular beds, brains from P0, P5, P7, P14 and P21 *Aldh1l1-eGFP* (BAC) males were fixed in 4% paraformaldehyde (PFA) overnight at 4 °C. Cortices were embedded in agarose and cut into serial, 50 μm-thick, coronal sections using a vibratome (VT1000S, Leica), then processed free-floating. Sections were blocked using a solution containing 10% donkey serum, 0.5% Triton X-100 in 50 mM PBS ('0.5% PBT') and 0.5% cold water fish skin

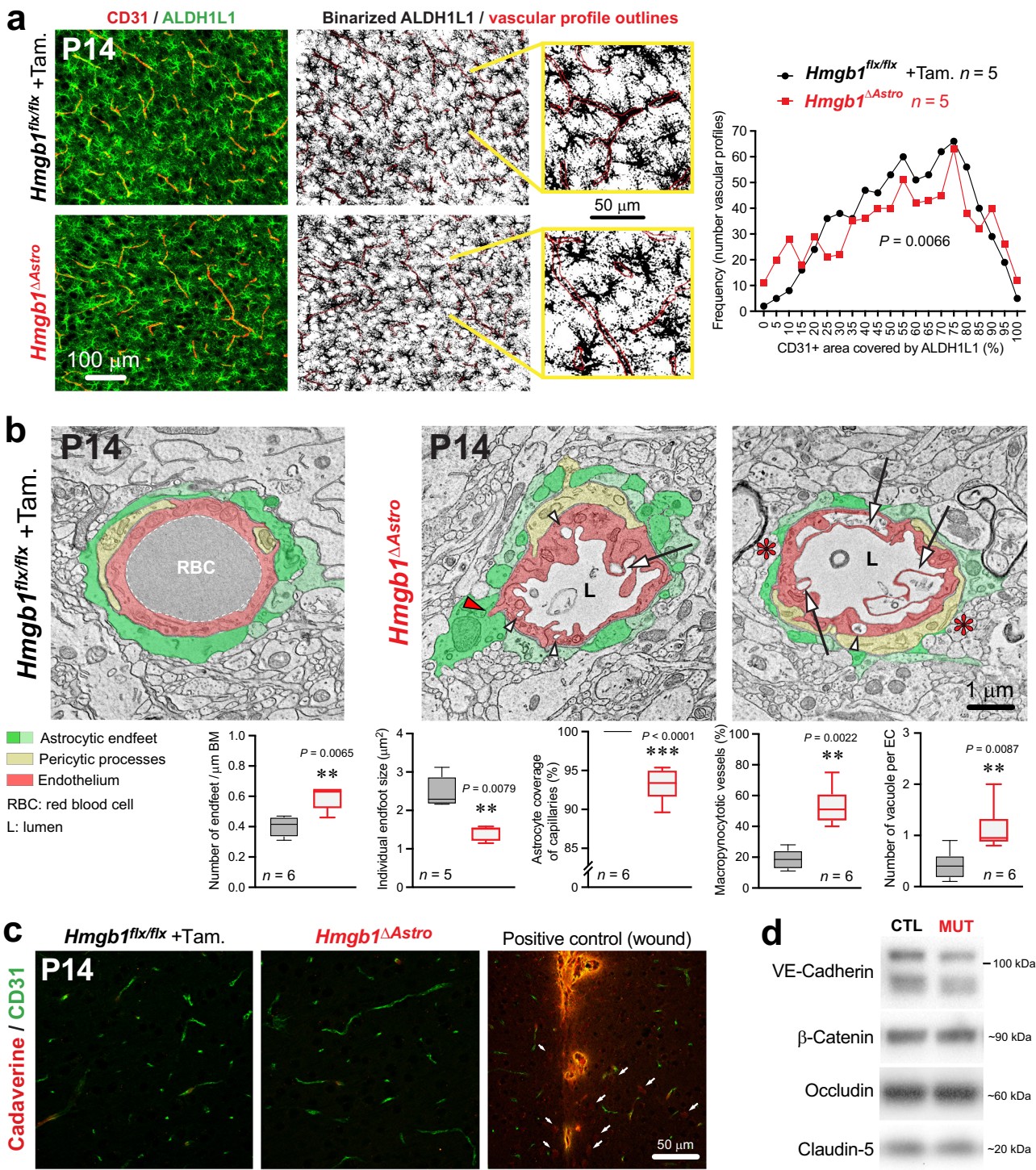

gelatin, then incubated overnight at room temperature (RT) with combinations of primary antibodies including: anti-CD31 (1:200; BD pharmingen Cat #553370), anti-GFP (1:50; Invitrogen Cat #ab5450) and anti-GFAP (1:500; Synaptic Systems Cat #173004). On the next day, sections were washed with 0.5% PBT and incubated with species-specific secondary antibodies (Goat anti-Guinea Pig Alexa Fluor488; Donkey anti-Rat Alexa Fluor568; Donkey anti-Goat Alexa Fluor488, Invitrogen) at 1:300 dilution for 3 h at RT. Sections were washed with 0.5% PBT, then 0.1 M PB (floating sections mounted) then coverslipped in Fluoromount-G (Electron Microscopy Science, EMS). Immunostained sections were examined under a Zeiss Axio Imager M2 microscope equipped with a Axiocam 506 Mono digital camera and the Zeiss

ApoTome.2 module for optical sectioning, controlled by ZenPro 2.6 software. Images were acquired as follows: 20 μm deep z-stacks were acquired with a 20x objective in the somatosensory area of the cortex and subsequently transformed into maximal intensity projections for 2D quantifications (see "Methods for automated analysis").

For immunostaining of brain sections from $Hmgb1^{flx/flx}$ and $Hmgb1^{\Delta Astro}$ mice, following overnight fixation in 4% PFA all brains were rinsed in PBS and submerged in 30% sucrose until tissue fully sank, then embedded in OCT medium and cut coronally into 25 μm-thick serial sections using a cryostat (HM525 NX, Thermo Fisher Scientific), then mounted on charged glass slides. Immunostaining was performed as previously described[24]. In brief, sections were incubated in a

**Fig. 4 | HMGB1 ablation in newborn astrocytes perturbs endfoot placement and endothelial morphology. a** Left, Fluorescence micrographs of immunostained cortical sections (ALDH1L1 green, CD31 red) from P14 control and *Hmgb1^ΔΔAstro^* mice. Middle: Binarized images of ALDH1L1 staining with superimposed CD31 mask (red outlines). Higher magnification images illustrating detail of astrocyte endfeet coverage on CD31 ROI are displayed in yellow lines. Right: Distribution graph reveals significant differences in the frequency distribution of CD31 coverage by ALDH1L1 immunoreactivity between control and *Hmgb1^ΔΔAstro^* mice (**$p < 0.01$, two-tailed Kolmogorov-Smirnov test). **b** Electron micrographs illustrating ultra-structural differences between control and *Hmgb1^ΔΔAstro^* mice in astroglial endfeet (pseudocoloured in two shades of green for better distinction between individual endfeet) and endothelial cells (pseudocoloured in light red). Pericyte processes were pseudocoloured in light yellow. White arrows indicate macropynocytotic extensions; white arrowheads indicate macropinocytotic vacuoles; red arrowheads point endothelial protrusions extending into astroglial endfeet; red asterisks indicate regions of vessel wall lacking endfoot coverage. Graph display quantifications of the average number of endfeet /μm basement membrane (BM); average endfoot size (μm²); average proportion of macropynocytic vessels, and average number of vacuoles per endothelial cells (EC). L lumen, RBC red blood cell. Data are whisker boxes (min to max, center line indicating median). **$p < 0.01$ (two-tailed Mann–Whitney's test). **c** Fluorescence micrographs of cortical sections immunostained for CD31 (green) and cadaverine Alexa Fluor-555 (red) in P14 *Hmgb1^flx/flx^* (left) and *Hmgb1^ΔΔAstro^* (middle) mice. Left, positive control: extravasation of cadaverine in the cerebral cortex 20 min after a stab wound with a 30 G needle. White arrows indicate cells that have taken up the dye. **d** Unchanged levels of inter-endothelial junction proteins VE-Cadherin, β-Catenin, Claudin-5, and Occludin as revealed by western blot from cerebral cortex extracts at P14. CTL control (*Hmgb1^flx/flx^*), MUT mutant (*Hmgb1^ΔΔAstro^*). All microscopy images displayed in this figure are representative of experiments repeated in at least 5 mice per group, with similar results. Source data are provided as a Source Data file.

blocking solution (2 h RT), followed by overnight incubation at RT with combinations of primary antibodies including anti-CD31 (1:300, BD pharmingen), -HMGB1 (1:1000, Abcam, Cat# ab18256), -Cx43 (1:400, MilliporeSigma, Cat# 6219), -Cre (1:1000, MilliporeSigma, Cat# 69050), -IBA1 (1:300, Abcam, Cat# ab107159), -NeuN (1:1500, Millipore Sigma, Cat# ABN90), -Ctip2 (1:200, Abcam, Cat# ab18465), -Pou3f2 (1:200, Cell Signaling Cat# 12137 S), -Ki67 (1:250, Thermo Fisher Scientific, Cat# MA5-14520), -phospho-histone H3 (1:300, Cell Signaling, Cat# 3377 S), -c-Fos (1:1000, Abcam, Cat# ab190289) -GFAP (1:500, SynapticSystems), -Aquaporin-4 (1:200, Alomone Labs, Cat# AQP-004) as well as with Click-iT™ EdU Cell Proliferation Kit (Thermo Fisher Scientific) according to manufactures protocol. For antibodies goat anti-Aldh1l1 (1:100, Thermo Fisher Scientific, Cat# 600-101-HB6) and monoclonal rabbit anti-ALDH1L1 (1:400, Thermo Fisher Scientific, Cat# 702573), a mild antigen retrieval step was performed. In brief, sections were incubated in 10 mM sodium citrate and 0.05% Tween-20 solution at 55 °C for 5 min and sections were then processed for immunofluorescence as outlined above. On the next day, sections were washed with 0.5% PBT and incubated with species-specific (Goat anti-Guinea pig 594 or 488, Donkey anti-Rat 568 or 488, -Rabbit 647 or 568, -Goat 488, -Mouse 568) AlexaFluor secondary antibodies (1:400, Invitrogen) for 2 h at RT. Sections were counter stained with DAPI then washed with 0.5% PBT and PB prior to coverslipping in Fluoromount-G (EMS). Immunostained sections were examined under the Zeiss Axio Imager M2 with ApoTome.2. A 20x objective was used to aquire 20 μm deep z-stacks in the somatosensory area of the cerebral cortex and subsequently transformed into maximal intensity projections for 2D quantifications (see "Methods for automated analysis"). Postnatal proliferation of endothelial cells and astrocytes were quantified as described previously[72].

For immunostaining of *Hmgb1^flx/flx^* and *Hmgb1^ΔΔAstro^* astrocyte cell cultures grown on 0.01% poly-l-lysine coated cover slips, cells were fixed with cold 4% PFA for 5 min then washed twice with PBS. Cells were then incubated with blocking solution for 1 h at RT. Cells were then incubated with primary antibodies anti-HMGB1 (1:1000, Abcam), -GFAP (1:1000, SynapticSystems), or - Phalloidin (1:40, ThermoFisher, Cat# A12379) for 2 h at RT followed by 3 washes with 0.2% PBT. Cell were incubated with species-specific (Donkey anti-Rabbit 647, Goat anti-Guinea Pig 568) AlexaFluor secondary antibodies (1:300, Invitrogen) for 1 h at RT. Cover slips were counter stained with DAPI then washed with 0.2% PBT and PB prior to coverslipping in Fluoromount-G (EMS) and allowed to dry. Cells were examined under the Zeiss Axio Imager M2 with ApoTome.2. A 20x objective was used to acquire images for morphological analysis. The proportion (% total number) of DAPI-positive astrocytes displaying long, thin extensions, or an acting ring, was measured manually, each cell contour being defined by GFAP immunoreactivity. The GFAP-positive cellular area was measured by manual tracing (*n* = 20 astrocytes per animal, *n* = 6 animals per group, from

3 experimental replicates) in ImageJ software (NIH) using the Segmented Line tool and Measure function on calibrated micrographs.

**Immunofluorescence for three-dimensional (3D) imaging.** Flatten-fixed cortices from *Hmgb1^flx/flx^* and *Hmgb1^ΔΔAstro^* P14 male and female mice were embedded in agarose and cut tangentially, through the somatosensory cortex, into serial 120 μm-thick sections with a vibratome and processed free-floating, as described above. Sections were incubated overnight at RT with anti-CD31 (1:200) as previously described[24]. The next day, sections were washed with 0.5% PBT and incubated with species-specific (Donkey anti-Rat 488) AlexaFluor secondary antibodies (1:300, Invitrogen) for 3 h at RT. Sections were washed with 0.5% PBT and 0.1 M phosphate buffer (PB), mounted on charged slides, then coverslipped in Fluoromount G (EMS). Immunostained sections were examined using a Zeiss Axio Imager M2 microscope equipped with a digital camera (Axiocam 506 mono) and the Zeiss ApoTome.2 module for optical sectioning. Nine 50-μm-deep z-stacks were then acquired at 10× objective in three major subdivisions of the cortex (anterior, parietal and occipital) for representative sampling of the cerebral cortex. The parietal subdivision corresponded to the primary somatosensory cortex. Tangential sections above and below layer IV were considered as layer II/III and V, respectively, as previously described[8,24]. Computational morphometric analysis of 3D vascular images was performed using a method we previously developed[8] (see also "Methods for automated analysis").

**Automated analysis of immunofluorescence images**
The diagram below presents the overall structure of the automated image processing and analysis procedures adopted for the present work, which incorporates five main blocks respectively analyzing: (a) astrocytes, (b) blood vessels, (c) connexin43, (d) neurons and (e) the quantification of the relationship between astrocytes, blood vessels and connexin43 (Supplementary Fig. 11). The main steps involved in the astrocytes identification and characterization are shown in (a). The analysis is divided into two main branches aimed at identifying: (a1) astrocyte body; and (a2) astrocyte processes. First, a Convolutional Neural Network (CNN) was used for segmenting astrocytes bodies, as indicated in branch (a1). The dataset consists of 238 images with size 1376 × 1104 pixels containing fluorescently labeled astrocytes. For training the network, 125 samples with size 384 × 384 pixels were extracted from the images. The images and locations used for extraction were randomly selected, with the restriction that no image had more than one sample extracted and that 25 samples were chosen for each age (P0, P5, P7, P14 and P21). All astrocytes bodies were manually marked on the 125 extracted samples. 80% of the samples for each age were used for training the network and 20% were used for validation. A U-Net architecture[90] with a ResNet-18 encoder[91] was trained for 100 epochs using the Adam optimization algorithm[92], a Dice loss[93] and the one cycle policy learning rate[94] with a maximum

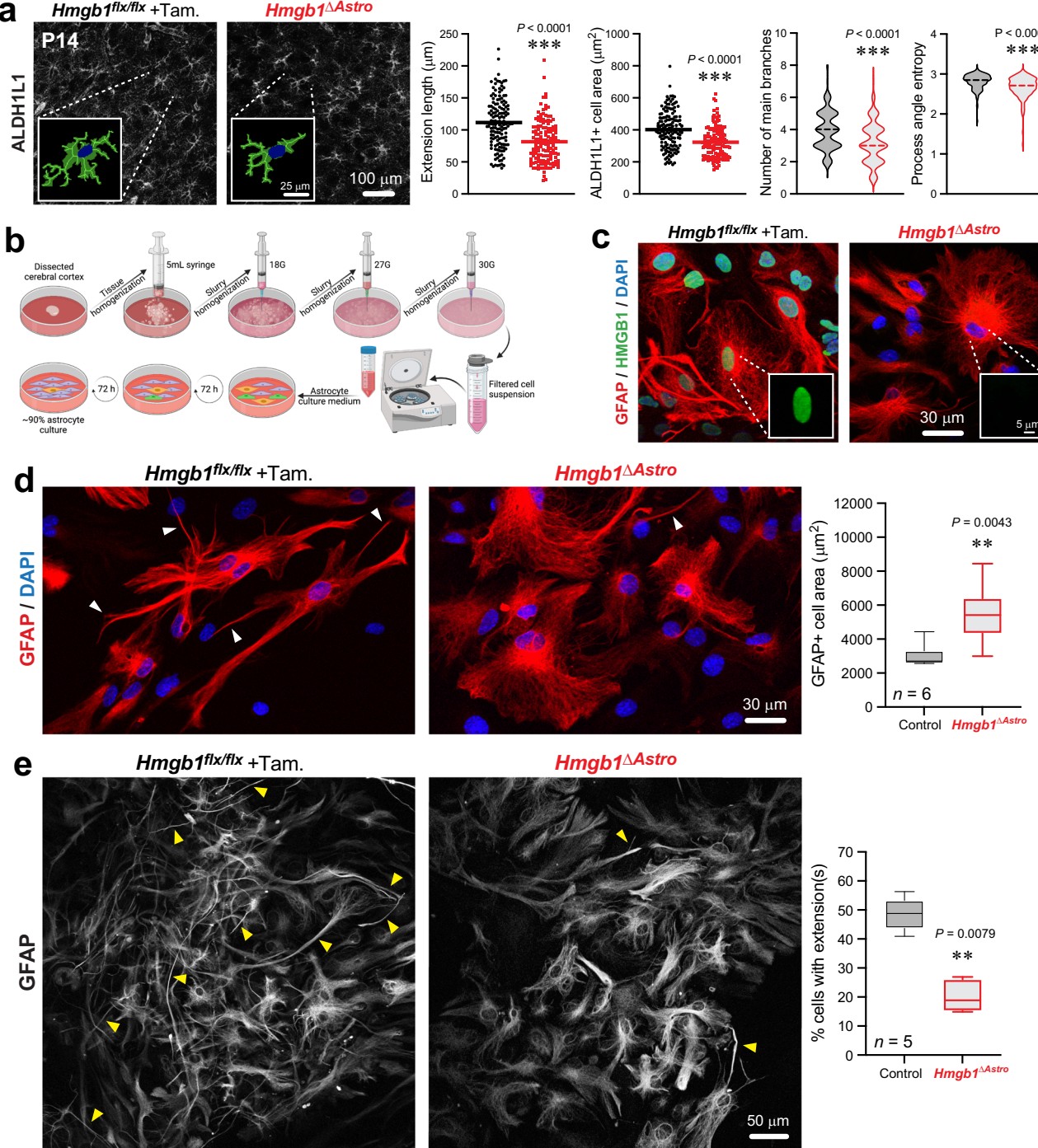

**Fig. 5 | Lack of HMGB1 in newborn astrocytes affects astroglial morphology in vivo and in vitro. a** Left, Fluorescence micrographs of immunostained cortical sections (ALDH1L1, desaturated) from P14 control and *Hmgb1ΔAstro* mice. Inset represent high magnifications of astrocytes segmented for main branches. *Right*, Quantification of morphological metrics from all astrocytes segmented for the control group (*n* = 149 cells from 4 animals) and for the mutant group (*n* = 143 cells from 4 animals). See Extended Data Fig. 7. for all cells. Data are mean with individual values (extension length, cell area) or violin plots (center dashed line indicating median). ***$p < 0.01$ (two-tailed unpaired *t* test). **b** Summary of procedure used to isolate and culture primary cerebral cortex astrocytes. **c** Fluorescence micrographs of immunostained primary astrocytes (GFAP, red; HMGB1, green; and DNA stain DAPI, blue) cultured after isolation from P14 control and *Hmgb1ΔAstro* mice. Absence of HMGB1 can be confirmed in *Hmgb1ΔAstro* astrocytes. **d** Left, Representative

fluorescence micrographs of immunostained primary astrocytes (GFAP, red; and DNA stain DAPI, blue) cultured after isolation from P14 control and *Hmgb1ΔAstro* mice. White arrowheads point at thin cellular extensions. Right, Graph shows quantification of GFAP+ cell area in culture. Data are whisker boxes (min to max, center line indicating median). ***$p < 0.01$ (two-tailed Mann−Whitney's test). **e** Left, Fluorescence micrographs of immunostained primary astrocytes (GFAP, desaturated) cultured after isolation from P14 control and *Hmgb1ΔAstro* mice. Yellow arrowheads point at thin cellular extensions. Right, Graph shows quantification of the proportion of astrocytes (% cells) displaying thin cellular extensions. Data are whisker boxes (min to max, center line indicating median). ***$p < 0.01$ (two-tailed Mann−Whitney's test). All microscopy images displayed in this figure are representative of experiments repeated in at least 5 mice per group, with similar results. Source data are provided as a Source Data file.

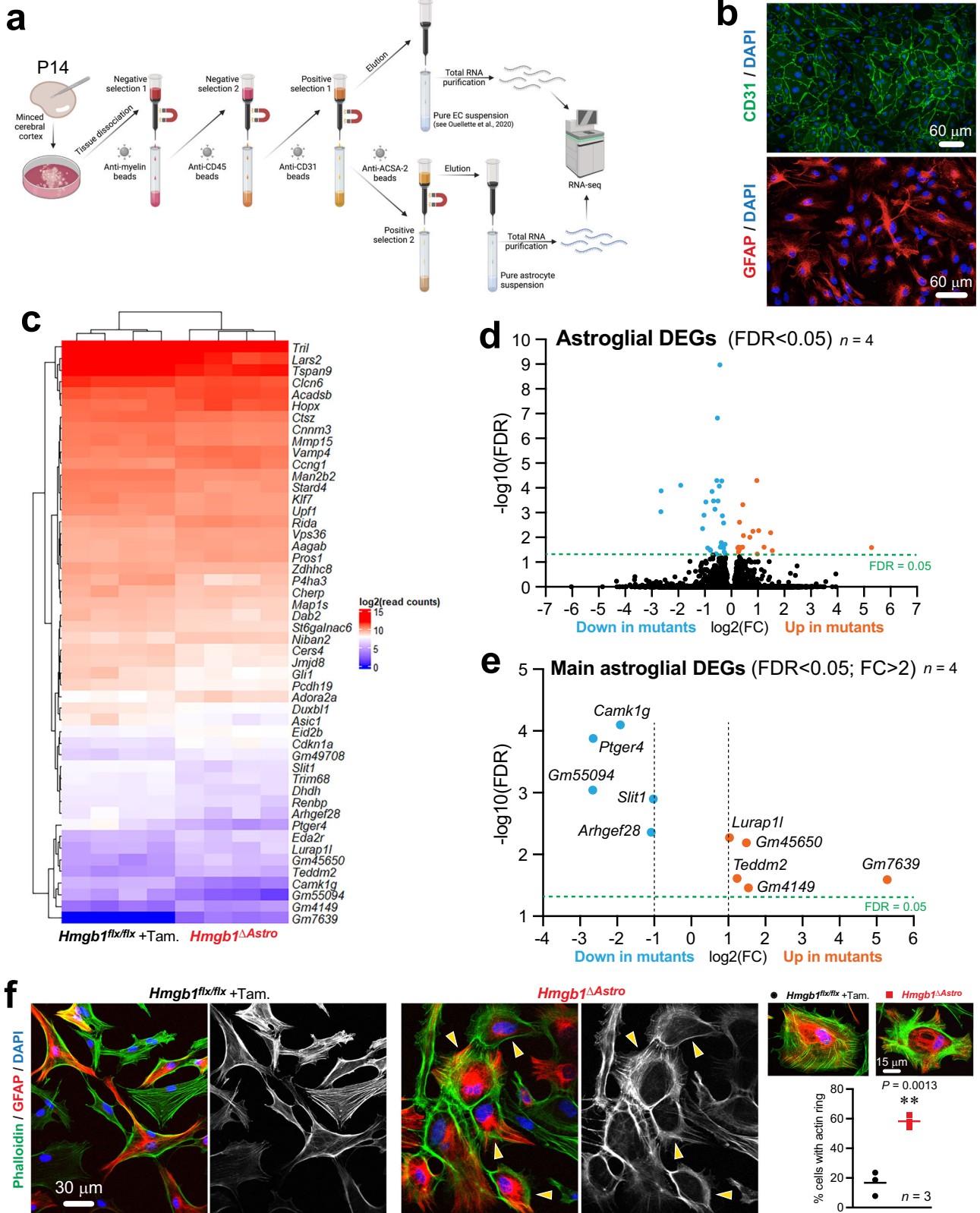

learning rate of 0.001. The following data augmentations were used for increasing the generalizability of the manual annotations: Gaussian blur, random gamma changes, Gaussian noise, left-right and up-down flips and random crops with resize.

Astrocytes processes identification, corresponding to the branch (a2), involved the following procedure. First, a Gaussian smoothing filter was applied. Next, the processes were segmented using an adaptive local thresholding technique. Each pixel was classified as an astrocyte process if its value was larger than the average intensity of a circular window of radius 20 μm centered on the pixel. Components with size smaller than 25 μm² were removed from the images. Astrocytes bodies identified using the aforementioned procedure were added to the segmentations. In order to characterize the processes, their medial lines were calculated using the Palágyi-Kuba thinning

**Fig. 6 | HMGB1 regulates a cytoskeleton-related genetic program in astrocytes.** **a** Summary of procedure to acutely purify total endothelial and astroglial RNA for deep sequencing analysis. Following microdissection and dissociation of the cerebral cortex, series of selections (negative followed by positive) allow for isolation of pure populations of ECs or astrocytes (see methods for details). **b** Sample images of immunocytochemical staining for endothelial marker CD31 or astroglial marker GFAP in primary culture performed after cell isolation steps for deep RNA sequencing, controlling the purity of the preparation. **c** Heat map illustrating differentially-expressed genes (False discovery rate, FDR < 0.05) between primary astrocytes from control and *Hmgb1^ΔAstro* mice (*n* = 4 group). **d**, **e** Volcano plots (FDR versus Log2 fold change, FC) display astroglial genes differentially-expressed between control and *Hmgb1^ΔAstro* mice. **f** Left, Fluorescence micrographs showing phalloidin staining (F-actin, green) of GFAP-positive astrocytes (red) in primary cell cultures obtained from control or *Hmgb1^ΔAstro* mice at P14. Yellow arrowheads point at F-actin ring-like structures in mutant astrocytes. Right, Higher magnifications of primary astrocyte cultures stained as in *left* to illustrate the remodeling of F-actin networks into a somatic ring in *Hmgb1^ΔAstro* astrocytes, with quantification (astrocytes from *n* = 3 animals per genotype). All microscopy images displayed in this figure are representative of experiments repeated in at least 3 mice per group, with similar results. Source data are provided as a Source Data file.

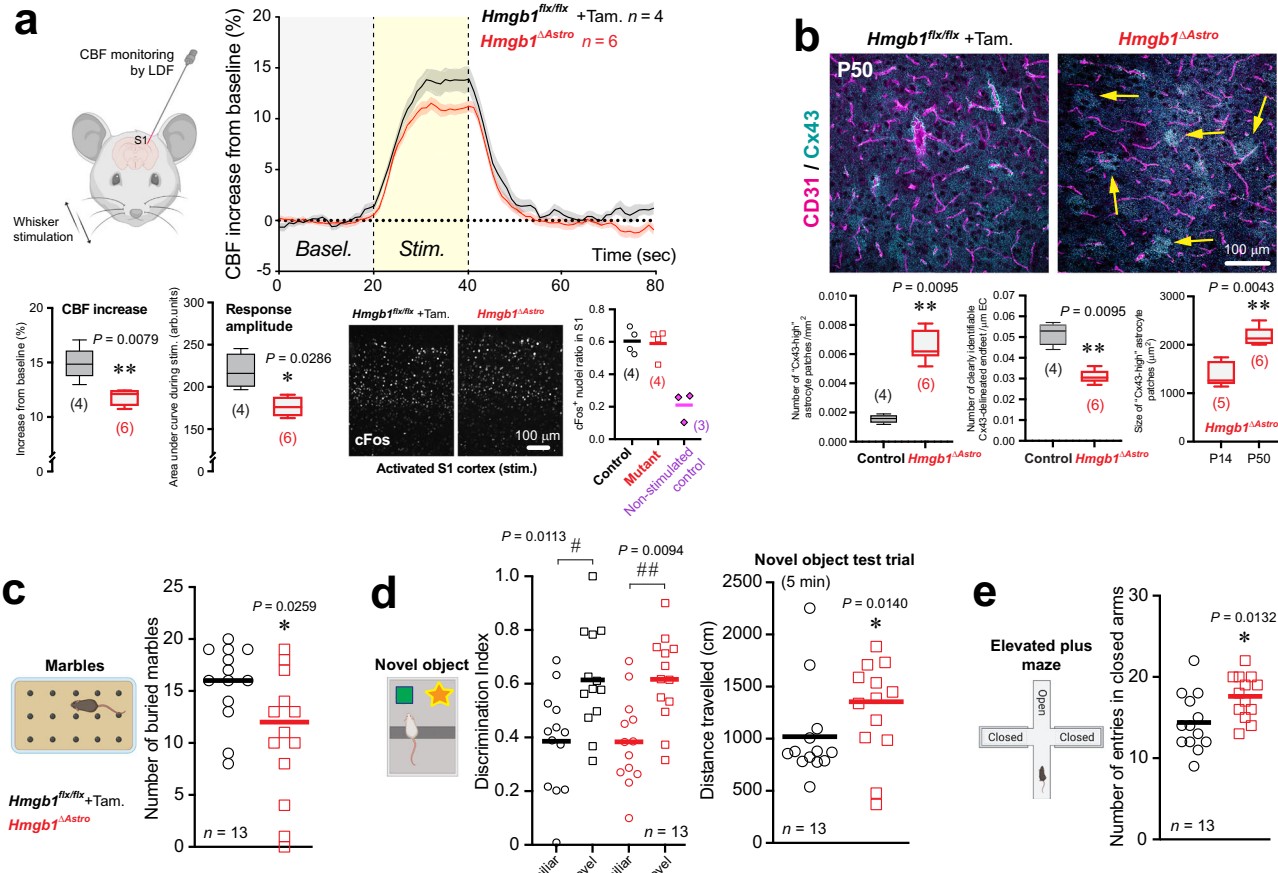

**Fig. 7 | Astroglial HMGB1 is required in newborn astrocytes for normal neurovascular coupling and behavior in adult mice. a** Top left: Schematic illustrating Laser Doppler flowmetry (LDF) coupled to whisker stimulation in P50 mice. Top right: Traces of normalized & averaged cerebral blood flow (CBF) responses. Traces are mean ± SEM. *Bottom*: Altered hemodynamic parameters from evoked CBF in the primary somatosensory (S1) cortex, and cFos immunostaining in S1. Data are whisker boxes (min to max, center line indicating median). \*p < 0.05, \*\*p < 0.01 (two-tailed Mann–Whitney's test). **b** Left: Fluorescence micrograph showing normal Cx43 (cyan) distribution around CD31-labelled microvessels (magenta) in cortical brain sections (S1) from control *Hmgb1^flx/flx* mice at P50. Right: Representative micrographs showing disruption of Cx43 distribution in cortical brain sections (S1) from age-matched *Hmgb1^ΔAstro* mice. Yellow arrows indicate "Cx43-high" patches in adult *Hmgb1^ΔAstro* mice. Quantification of Cx43 distribution demonstrates that the number of "Cx43-high" patches remains significantly higher in *Hmgb1^ΔAstro* mice compared to controls at P50. Consistent with *Hmgb1^ΔAstro* mice at P14, the number of clearly identifiable Cx43-delineated endfeet is reduced in adults *Hmgb1^ΔAstro* mice. However, the size of "Cx43-high" patches in *Hmgb1^ΔAstro* mice is significantly higher at P50 than at P14. Data are whisker boxes (min to max, center line indicating median). \*\*p < 0.01 (unpaired two-tailed Mann–Whitney's test). **c** Behavioral assessment measured decreased marble burying in *Hmgb1^ΔAstro* mice compared with control littermates. **d** In the novel object recognition task, *Hmgb1^ΔAstro* mice travelled more distance during the 5-min trial. **e** In the elevated plus maze, *Hmgb1^ΔAstro* mice displayed increased entries in closed arms. Schematic representations are displayed for each behavioral task. All behavioral data are mean with individual values. #p < 0.05, ##p < 0.01 (Two-way ANOVA and Sidak's post-hoc test). \*p < 0.05 (two-tailed Mann–Whitney's test). All microscopy images displayed in this figure are representative of experiments repeated in at least 4 mice per group, with similar results. Source data are provided as a Source Data file.

algorithm[95]. Pixels belonging to a medial line and having three or more neighboring medial line pixels were classified as bifurcations, while pixels in a medial line having one neighbor were classified as terminations. Each astrocyte was then stored as a graph, where a root node represented the cell body and the remaining nodes represented bifurcations and terminations of the astrocytes processes. Edges with

size smaller than 4 μm were considered spurious and were removed from the graphs.

The astrocyte density in a sample was calculated as the number of astrocytes divided by the area of the sample. The astrocytes extension length was calculated as the sum of the arc-lengths of the medial lines of the identified astrocytes processes divided by the area of the

sample. The extension branching point density was calculated as the number of nodes having degree larger than or equal to 3 in the graph representation of the processes divided by the area of the sample.

The astrocyte process angle entropy was calculated as follows. For each point in the medial line of an astrocyte process, henceforth referred to as a reference point, all points in the same medial line that were at a distance smaller than 4.5 μm from the reference point were identified. These points defined a portion of the medial line segment around the reference point. A straight line was adjusted to the segment, and the angle of the line was associated with the reference point. This process was repeated for all points in the medial lines. A histogram was constructed from all angles calculated for a given astrocyte, and the histogram values were used for calculating the Shannon entropy[96] of the angles. A low entropy of the angles means that most extensions of the astrocyte tend to have the same angle, while a high entropy indicates that there is no preferred angle. The overall angle entropy of a sample was calculated as the average angle entropy of the astrocytes.

The main procedures used for analyzing the blood vessels are shown in (b). The blood vessels were segmented using a CNN that was trained on segmented images obtained from previously published works[8,24]. The network architecture was similar to the one used for segmenting the astrocytes, with the difference that the training was done for 10 epochs and used a maximum learning rate of 0.01. The transformations for data augmentation were also similar, with the addition that histogram equalization was applied to the images. The medial lines of the detected blood vessels were calculated, and a graph was used for representing bifurcations and terminations of the vessels, similarly to what was done for the astrocytic extensions.

The blood vessel density was calculated as the arc-length of all medial lines of the blood vessels in a sample divided by the sample area. The branching point density was calculated as the number of nodes with degree equal to or larger than 3 in the graph divided by the sample area.

The connexin was segmented as follows (c). First, a Gaussian blur was applied to the images. Next, an adaptive local thresholding operation was applied for classifying the pixels into connexin or background. Pixels having a value larger than the average value of a circular window of radius 20 μm around them were classified as connexin. Components with an area smaller than 15 μm² were understood as background.

The neurons were identified as follows (d). First, a histogram equalization procedure was applied to the images. Neurons in the images could be regarded as bright blobs on a dark background. Thus, a multi-scale Laplacian of Gaussian (LoG) blob detector[97] was used. The scales used for the filter (standard deviation of the Gaussians) ranged from 4.5 μm to 9 μm. Peaks in the scale-space volume generated by the LoG filters with value larger than 0.1 were associated with individual neurons.

Neuron density along cortical depth was calculated as follows. The cortices surfaces were marked manually and used as a reference for calculating a distance transform[98] from the surface. The calculated distances were then divided into equally spaced 50 μm thick areas, and the neuron density was calculated for each section.

Additional procedures were defined for quantifying the relationship between different structures, as illustrated in (e). Astrocytes containing processes that overlapped with blood vessels were identified using the astrocytes and blood vessels segmentations. The fraction of astrocytes contacting blood vessels was calculated as the number of astrocytes with at least one process overlapping a blood vessel divided by the number of astrocytes in the sample. The segmentations were also used for calculating astrocyte cell body density as well as EGFP density as a function of distance from blood vessel wall. A set of regions of interest were defined at distance intervals of 1 μm from all vessel walls in a sample, and the astrocyte cell density and

EGFP density were calculated for each region. The densities were then normalized by the overall density of the respective sample.

Blood vessel connexin coverage was calculated using the connexin and blood vessels segmentations. Connexin having a distance equal to or smaller than 6 μm from any segmented blood vessel was considered to overlap with blood vessels. The total area of connexin overlapping with blood vessels divided by the total blood vessel area in a sample defined the connexin coverage. The coverage was calculated separately for blood vessels having diameter <=4.5 μm and those having diameter >4.5 μm.

### Tamoxifen-inducible Cre-recombinase activation

Tamoxifen injections were performed as previously described[99]. Briefly, for a stock solution 10 mg of tamoxifen (Sigma) were dissolved in 250 μl of 100% (vol/vol) ethanol and vortexed at maximum speed until tamoxifen was completely dissolved. The tamoxifen solution was then mixed with 750 μl of peanut oil (Sigma) and vortexed thoroughly for 15–20 min until an emulsion was obtained. The stock solution was diluted in a ratio of 1:10 with peanut oil and vortexed thoroughly for 15–20 min, for injections. Three consecutive injections of tamoxifen (50 μl, 1 mg/ml) into the mouse's stomach was performed at P0, P1 and P2 in all pups from breeders (*Hmgb1*<sup>flox/flox</sup> x *Aldh1l1-CreER*<sup>T2+/−</sup>;*Hmgb1*<sup>flx/flx</sup>). All injections were carried out in the same way at the same hour and the solution was always well homogenized before injecting.

### Laser Doppler flowmetry (LDF)

LDF measurements of evoked cerebral blood flow (CBF) were performed using the BLF22 flowmeter and a needle probe (Transonic Systems Inc., Ithica, NY, USA) in response to sensory stimulation in anesthetized mice (ketamine 100 mg/kg intraperitoneally; Bimeda, Cambridge, ON, Canada), as previously described[24]. In brief, CBF was recorded over the left somatosensory cortex before, during and after unilateral stimulation of the right whiskers with an electric toothbrush (20 s at 8–10 Hz). Recordings from 4 to 6 stimulations (every 60 s) were acquired, using LabChart 8 software, and averaged for each mouse. Quantifications of hemodynamic parameters (% increase from baseline, rising slope, time to maximum response, falling slope) were inspired from a reference hemodynamic study using multispectral optical intrinsic signal imaging[100].

### Blood-brain barrier (BBB) permeability analysis

Experiments to assess BBB integrity were performed according to[11]. Briefly, 50 μl of Alexa Fluor 555 cadaverine (1 mg/ml, Thermo Fisher Scientific) were injected intraperitoneally in P14 mice. Cadaverine was allowed to circulate for 16 h, and mice were then perfused intracardially with cold PBS. Brains were extracted from *Hmgb1*<sup>flx/flx</sup> and *Hmgb1*<sup>ΔAstro</sup> animals, fixed in 4% PFA at 4 °C overnight, and further processed as described above for 2D immunofluorescence for visualization of cadaverine extravasation.

### Western blot analysis

Protein samples from *Hmgb1*<sup>flx/flx</sup> and *Hmgb1*<sup>ΔAstro</sup> cortical tissue were isolated using RIPA buffer with protease and phosphatase inhibitors and resolved by sodium dodecyl-sulfate polyacrylamide gel electrophoresis (SDS-PAGE) as described previously[72]. Briefly, 30 μg protein was separated on a 10% SDS-PAGE gel (TGX Stain-Free FastCast, BioRad) and transferred to polyvinylidene difluoride membranes. Membranes were processed and incubated overnight at 4 °C with primary antibodies anti-HMGB1 (1:2000, Abcam); -Cx43 (1:5000, Sigma-Aldrich), -VE-Cadherin (1:1000, Abcam Cat# ab205336), -β-catenin (1:1000, Abcam Cat# ab32572), -Occludin (1:1000, Abcam Cat# ab216327), -Claudin-5 (1:1000, Invitrogen Cat# 34-1600), -Aquaporin-4 (1:5000, SynapticSystems Cat# 429011), -Aquaporin-4 (1:200, Alomone labs) or -γ-tubulin (1:3000, Thermo Fisher Scientific Cat# MA1-850) in Tris-buffered saline (Tris 50 mM and NaCl 150 mM, pH 8.0) containing

0.05% Tween (TBST) in 1% nonfat dry milk. The membranes were washed and incubated with horseradish peroxidase–conjugated secondary antibody: Anti-Rabbit HRP or Anti-mouse HRP (Promega or Thermo Fisher Scientific) 1:10,000 in TBST containing 5% nonfat dry milk. Immunoreactive proteins were visualized by chemiluminescent solution (SuperSignal West Dura; Pierce Biotechnology). Densitometric, semiquantitative analysis of Western blots was performed using Image Lab software and normalized to total protein.

### Transmission electron microscopy (TEM)

Mice were anesthetized with a ketamine/xylazine cocktail (1 mg/kg, intraperitoneally) and perfused through the aortic arch with 3.5% acrolein and 4% paraformaldehyde. Fifty-micrometer-thick coronal sections of the brains were cut in sodium phosphate buffer (PBS 50 mM, pH7.4) using a Leica VT1000S vibratome (Leica Biosystems) and stored at −20 °C in cryoprotectant until further processing. Sections were post-fixed and embedded using variations of the protocol by Deerinck et al. (https://ncmir.ucsd.edu/sbem-protocol). In brief, the sections were washed 3 times in PBS for 10 min and were incubated in 1.5% potassium ferrocyanide and 2% aqueous osmium tetroxide in 0.1 M phosphate buffer (pH 7.4) for 1 h at room temperature. The sections were subsequently washed 5 times with double-distilled water (ddH$_2$O) for 3 min, then incubated 20 min in a fresh solution of thiocarbohydrazide (1% w/v) at room temperature. Sections were washed again 5 times with ddH$_2$O for 3 min, incubated 30 min in 2% aqueous osmium tetroxide, and washed 5 times with ddH$_2$O for 3 min. Sections were dehydrated using increasing ethanol concentrations followed by propylene oxide, and then embedded in Durcupan resin (Sigma-Aldrich) between ACLAR sheets at 55 °C for 3 days, as described previously[101]. Ultrathin sections were generated at ~65 nm using a Leica UC7 ultramicrotome. Imaging was performed in the antero-frontal (2.8 to 1.98 mm Bregma) and parieto-somatosensory (−0.70 to −1.82 mm Bregma) areas. In each region, 25 capillaries per animal were randomly photographed using a JEM-1400 (JEOL) transmission electron microscope operating at 60 kV and equipped with a 4k Gatan OneView camera, TEM center and DigitalMicrograph software were used to control TEM and camera, respectively. For each capillary, an image at 5500× and 10,000× was acquired, to generate a high-resolution mosaic of the capillary. Analyses of the micrographs were performed by an experimenter blind to the conditions using ImageJ software. Quantifications of endfoot number and coverage, as well as caveolae vesicle numbers were performed as described previously[24,102]. To analyze astrocyte endfeet, a region of interest was drawn around each astrocyte endfoot in contact with the blood vessel. Individual endfeet were defined as closed areas of dim, electron-clear astroglial cytoplasm in contact with the exterior aspect of the basement membrane (BM). As previously described[103], astrocyte endfeet are also characterized by presence of long, enlarged cisternae and electron-dense mitochondria. Neighbouring endfeet appeared characteristically separated by a clear membrane line (inter-endfoot wall). The Measure function was used to calculate the area of the each endfoot. For each image, the total endfoot number in contact with the vessel was tallied, as was the combined area of all endfeet. The length of the BM surrounding the endothelium, and of the astrocytic membrane in direct contact with the BM was measured to obtain the proportion of BM (i.e., capillary wall) covered by astrocyte endfeet. All metrics were quantified manually from scaled micrographs in ImageJ (NIH), using tools as Polygon, Freehand Selections or Straight Line, followed by the Measure function.

### Quantitative multiplexed digital characterization of mRNA targets in brain tissue sections

**Tissue preparation.** Brain tissue from P0, P5, P14, WT "Noncarrier" mice (from *Aldh1l1-eGFP* (BAC) strain) as well as P14 *Hmgb1^flx/flx* and *Hmgb1^ΔΔAstro* animals was extracted and fixed overnight at 4 °C in 4% RNase-free PFA (Electron Microscopy Sciences) in sterile PBS. Samples were then placed in 70% ethanol in sterile RNase-free water. Brain tissues were paraffin-embedded and sectioned (5-μm-thick) at the Louise Pelletier Histology Core Facility at Ottawa University, under RNase free conditions. Brain sections were mounted on Leica Bond Plus slides (Cat# S21.2113.A). After mounting, sections air-dried at room temperature for >30 min. Once dry, sections were stored at −80 °C in desiccator and shipped to NanoString Technologies (Seattle, WA) with dry ice along with rabbit anti-Aldh1L1 (Thermo Fisher Scientific) antibody.

**GeoMx™ DSP mouse WTA profiling.** 4 FFPE slides containing 8 samples (2 tissues per sample) were processed following the GeoMx® DSP slide prep user manual (MAN-10087-04). Before being deparaffinized and hydrated by Leica Biosystems BOND RX, the slides were baked in oven at 67 °C for at least 3 h. 0.1 ug/ml proteinase K was added to digest the proteins for 15 min before the mouse WTA probe mix was added on the slides for overnight hybridization. On the second day, the slides were washed with buffer and stained with mAldh1l1 (ThermoFisher, 702573) for 2 h, and then secondary anti-rabbit AF594 antibody was added on the tissue for 1 h. After 3 times of washing the slides, CD31 (R&D system, AF3628), and Syto83 (ThermoFisher, S11364) were added on the slides for 2 h at room temperature. Regions of interest (ROIs) were placed on 20× fluorescent images scanned by GeoMx™ DSP. Oligoes from selected regions were collected through UV-cleavage and transferred to 96-well plate. The oligoes then were uniquely indexed using Illumina's i5 × i7 dual-indexing system. PCR reactions were purified and libraries were paired-end sequenced (2 × 75) on a Novaseq instrument following the GeoMx® NGS library prep user manual (MAN-10117-05). Fastq files were further process by DND system and raw and Q3 normalized counts of all WTA targets in each AOI were obtained through GeoMx™ DSP data analysis software.

**GeoMx™ DSP mouse WTA data normalization and analysis.** GeoMx™ DSP counts from each AOI were analyzed following NanoString GeoMx Data Analysis User Manual, using the online version of the Nanostring GeoMx DSP Data Analysis Suite (v2.4) [SEV-00090-05]. Data were QC'd using QC parameters (min 1000 raw reads; min 80% aligned reads; no template PCR control (NTC) count <1000). Probes with expression levels above the default limit of quantitation (LOQ) in at least 10% of the samples, within a specific contrast, were retained for analysis. For each area of interest (AOI), the expected background was estimated as the geometric mean of the negative control probes, multiplied by the geometric standard deviation to the second power ($\wedge$2). Raw counts from each AOI were scaled to have the same 75th percentile of expression. The AOIs were categorized according to cell enrichment group, i.e., astrocyte-enriched (i.e., ALDH1L1 + CD31-) and microvessel-enriched (i.e., CD31 + ALDH1L1-). Within each cell enrichment group, differential expressions of genes between P0, P5 and P14 were analyzed using Linear Mixed Model (LMM) with Scan_ID as variable. The LMM was chosen to account for the repeated measurement (i.e. pseudo-replicates) in each mouse. Nominal $p$-values were FDR corrected to account for multiple tests, and genes with FDR < 0.05 were considered significant.

**Quantification of trends.** To measure genes expression level of over time, Q3 normalized counts for genes (targets) of interest were average per animal (i.e., average of two AOIs per mouse, n = mouse). Trends were calculated for selected genes from astrocyte-enriched (i.e., ALDH1L1 + CD31-) and microvessel-enriched (i.e. CD31 + ALDH1L1-) AOIs.

## Primary mouse brain astrocyte isolation for immunocytochemistry

All mice were euthanized by cervical dislocation. The cortex from either *Hmgb1flx/flx* or *Hmgb1ΔAstro* mouse was dissected in cold DMEM high glucose using autoclaved tools submerged in 100% ethanol 30 min prior dissection. Cortices were homogenized in DMEM high glucose using a 5 mL syringe and further dissociated with a (i) 18 G needle, (ii) 27 G needle and (iii) 30 G needle. The cell suspension was filtered using a 70 uM strainer and subsequently centrifuged at 300xg for 10 min. Pelleted cells were resuspended in astrocyte culture medium (DMEM high glucose (Cytiva, SH30243.01), 10% FBS (Wisent Bio-Products, 115727) and 1% penicillin/streptomycin (ThermoFisher, 15140122) and seeded on 6-well plates (one brain/well) coated with 0.01% poly-L-lysine. Astrocyte culture medium was replaced 72 h post-seeding and every 72 h until an appropriate level of confluence was reached. Cells were cultured for 1 week then passaged, on to cover slips for 3 days prior to fixation and immunostaining (see section Immunohistochemistry and immunofluorescence).

## Primary mouse brain endothelial cells and astrocytes isolation for bulk deep RNA-seq

All mice were euthanized by cervical dislocation. The cortex from either *Hmgb1flx/flx* or *Hmgb1ΔAstro* mouse was dissected in cold HBSS without calcium and magnesium using autoclaved tools submerged in 100% ethanol 30 min prior dissection. Cerebral cortex was minced to small 2–3 mm pieces and dissociated using Neural Tissue Dissociation Kit P compounds (Miltenyi Biotec, 130-092-628) to obtain a cell suspension. Cell isolation procedures were completed according to the manufacturer's instructions.

Cell suspension underwent two negative selection steps to deplete myelin (Myelin removal beads II, 130-096-733, Miltenyi Biotec) and CD45-positive cells (CD45 microbeads, 130-052-301, Miltenyi Biotec). Each depletion step consisted of incubating the cell suspension with magnetic beads (myelin removal or CD45 microbeads) which were then selected using a magnetic MACs separator. Following the negative selection, endothelial cells (ECs) were isolated by incubation with CD31-coated magnetic microbeads (130-097-418, Miltenyi Biotec) and the positive selection was performed using a magnetic MACs separator. Pure ECs were pelleted for subsequent acute RNA extraction. Astrocytes were isolated from the remaining cell suspension using the anti-ACSA-2 microbead kit (130-097-678, Miltenyi Biotec). The cell suspension was incubated with FcR blocking reagents followed by anti-ACSA-2 microbeads and placed on a magnetic MACS separator for positive selection. Pure astrocytes were pelleted for subsequent acute RNA extraction. RNA was extracted from endothelial cells and astrocytes by adding 300 μl of Trizol (ThermoFisher) to the cell pellet, then RNA was purified using the Direct-zol RNA Microprep (R2060, ZYMO RESAERCH), according to manufactures protocol, and eluted in 15 μl of DNase/RNase-free water.

## Bulk deep RNA-Seq of primary astrocytes and endothelial cells

RNA-Seq Libraries were generated using the NEBNext® Ultra™ II Directional RNA Library Prep Kit for Illumina (NEB). The RNA-Seq libraries were quantified by Qubit and qPCR according to the Illumina Sequencing Library qPCR Quantification Guide and the quality of the libraries was evaluated on Agilent Bioanalyzer 2100 using the Agilent DNA-1000 chip. The RNA-Seq library sequencing was performed using Illumina Next-Seq500. FASTQ file format were processed by trimming the adaptor sequences, filtering low-quality reads (Phred Score <= 20) and eliminating short reads (length <= 20 bps) using software package FASTX-toolkit [http://hannonlab.cshl.edu/fastx_toolkit/]. STAR (v2.7.9a)[104] was used for alignment of the reads to the reference genome and to generate gene-level read counts. Mouse (Mus musculus) reference genome (version GRCm39)[105] and corresponding annotation were used as reference for RNA-seq data alignment process. DESeq2[106]

was used for data normalization and differentially expressed gene identification for mutant versus wild-type astrocyte and endothelial cell respectively. Differentially expressed genes (DEGs) were defined by a False Discovery Rate (FDR)-adjusted $q$-value < 0.05. Main DEGs were defined by >2 fold of changes in ratio [abs(log2 fold-change) >1].

Two differential gene expression (DEG) analyses were performed on 16 samples, each sample (endothelial cell and astrocyte) extracted from one brain:

1. P14_Mut Endothelial cells ($n = 4$ mice) vs. P14_WT Endothelial cells ($n = 4$ mice)
2. P14_Mut Astrocytes ($n = 4$ mice) vs. P14_WT Astrocytes ($n = 4$ mice)

No sample was excluded. On average, 94% of the 24 million reads were aligned to Mouse (Mus musculus) reference genome.

## Mouse behavior assessment

Prior to behavioral testing, all animals were left to acclimatize to an reverse light cycle housing room for 10 days and handled once a day for 1 week. Behavioral tests were completed at University of Ottawa's Behavior Core Facility between 9am and 4 pm under dim red light, unless stated otherwise. On testing day, animals were habituated to the testing room for 30 min. Behavior tests were performed with all mice in the following order: marble burying test (1 day); novel object recognition (2 days); open field test (1 day). All tests were directly inspired or slightly modified from published studies[107,108].

**Novel object recognition test (NOR).** A 2-day NOR was as described elswhere[24,109]. On day one, habituation day, each animal was habituated to an empty arena (45 cm × 45 cm × 45 cm) for 30 min. On day two, experimental day, mice were habituated for a second time to the empty arena for 10 min. Following habituation, each mouse was removed from the empty arena and placed in a clean cage for 2 min. Two identical objects (red cup or white funnel) were placed in the arena and the mouse is returned to the arena for a 10 min familiarization period. The mouse was removed from the arena and placed in a clean cage for 1 h. Following this hour, the object recognition test consisted of one clean familiar object and one clean novel object (red cup or white funnel switched). The mouse was returned to the arena for a 5 min recognition period. All interactions with the objects were recorded using Ethovision 14 XT software (Noldus). Object recognition was scored as the time during which the nose of the animal was located within 2 cm of the object. A discrimination index was calculated as: [time spent interacting with novel object/ (time spent interacting with novel object + time spent interacting familiar object)].

**Marble burying test.** This test, performed as described elsewhere[24,108] consisted in a 30 min trial per mouse. For each trial, a standard polycarbonate rat cage (26 cm × 48 cm × 20 cm) is filled with 5 cm-thick SANI-chip bedding. Using a template, 20 standard blue glass marbles were gently placed (5 rows of 4 marbles) on the surface of the bedding. Each mouse was placed at the bottom left corner of the cage. During the test, the cage was covered by transparent Plexiglas. Each trial was recorded using Ethovision 14 XT software (Noldus). Once the trial was completed, the number of marbles that were either fully buried, or at least 2/3 covered by bedding, were counted as 'buried'. Between each trial, marbles were washed in mild detergent and dried.

**Open field test.** Anxiety-like behavior was examined using a 10 min open field test as described elsewhere[107]. This test was completed in bright light. Using a light meter, the light intensity was set to 300 lux for anxiety measurement. Each mouse was placed in the centre of an empty open field box (45 cm × 45 cm × 45 cm) and left to explore for 10 min. Three major arena zones were setup: 4 corner zones (each at 10 cm × 10 cm); large center zone (25 cm × 25 cm); small center zone

(15 cm × 15 cm). Total distance travelled (cm), velocity (cm/s) and time spent in each respective zones were recorded using Ethovision 14 XT software (Noldus).

**Elevated plus maze.** Standard test of anxiety-like behavior as described elsewhere[110]. Using a light meter, the light intensity was set to 100 lux. Each mouse was placed in the center of a maze composed of two arms crossed perpendicularly, each measuring 6 cm wide and 75 cm long. One arm consists of an open platform while the second arm is enclosed by 20 cm high walls. The mouse was left to explore the maze for 10 min. Ethovision 14 XT software (Noldus) was used to record the number of entries and time spent in open arms.

### Statistics and reproducibility

No statistical methods were used to pre-determine sample sizes, but our sample sizes are similar to those reported in previous publications[8,24,33,72,101]. All animal/sample/biological replicate numbers in this study are in line with well-accepted standards from the literature for each method. All data presented in this work were obtained from experimental replicates (e.g., multiple animal cohorts from different litters; at least three experimental repeats for each assay, multiple RNA extractions and production of biological replicates, etc.). All attempts of replication were successful. All data analysis was conducted blind to genotype/experimental condition. Randomization of individual samples/animals was performed by numbering. Groups were reassembled upon completion of data analysis according to genotype and age. All statistical tests were performed using GraphPad Prism 9.0 & 9.3 Software. Data distribution was assumed to be normal, but this was not formally tested. A two-tailed Mann–Whitney $U$ test, appropriate for small sample size (each animal being considered as a sample), was used for two-group comparisons. A one-way or two-way ANOVA, followed by a post-hoc test (Sidak's or Tukey's), was used for multigroup comparisons and trends. $p < 0.05$ was considered significant.

### Reporting summary

Further information on research design is available in the Nature Portfolio Reporting Summary linked to this article.

## Data availability

All data supporting the findings of this study are provided within the paper and in its Supplementary Information/Source data files. The RNA-sequencing data generated in this study have been deposited and are accessible through GEO via accession numbers GSE224784, GSE236429. The uncropped Western blots are provided in the Supplementary Information file. All protocols and raw data reported in this study are available from the corresponding author upon request. Source data are provided with this paper.

## Code availability

The code for automated analysis of immunofluorescence images has been deposited and is accessible through GitHub and from the Comin lab (chcomin@gmail.com).

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

## Acknowledgements

We thank Drs. Simon Chen and Chenghua Gu for their valuable comments on the manuscript; the histology services provided by the Louise Pelletier Histopathology Core Facility at the University of Ottawa; the animal care and veterinary services at the University of Ottawa; and Jinjing Gong, Katie Barret, Colleen Urban, Elizabeth Mahoney and Seth Meyers (NanoString Technologies, Inc.) for coordinating spatial transcriptomics experiments. We would also like to thank Cédric Gravel for

help with immunostaining work. For this work, B.L. was supported by research grants from the *Canadian Institutes of Health Research* (grant #388805) and the *Natural Sciences and Engineering Council of Canada* (grant RGPIN-2018-04456). M-E.T. was supported by grants from the *Natural Sciences and Engineering Council of Canada* (grant RGPIN- 2014-05308) and a *Tier 2 Canada Research Chair* in Neurobiology of Aging and Cognition. M.C. was supported by a Doctoral Scholarship from the *Fonds de recherche du Québec*. C.H.C. thanks *FAPESP* (grant #2021/12354-8). L.daF.C. thanks *CNPq* (grant #307085/2018-0) and *FAPESP* (grant #2015/22308-2).

## Author contributions

M.F.A., P.V.D., M.C., J.R.N., J.O., K.T., A.S., S.L., N.B. performed experiments. M.F.A., C.H.C., P.V.D., J.R.N., L.daF.C., Q.Y.L., Y.P., Z.L., and B.L. analyzed data/images. G.C and B.L. generated and/or analyzed spatial transcriptomic data. Q.Y.L., Y.P., Z.L., and B.L. generated and/or analyzed bulk RNAseq data. M.-E.T. supervised M.C. to provide expertise in TEM sample processing. B.L. conceived and led the project, designed experiments with M.F.A. and with input from G.M.R. and N.S. M.F.A. and B.L. prepared the manuscript and figures, with input from all co-authors.

## Competing interests

The authors declare no competing interests.
