## [Peer Review File · Nature Communications]

REVIEWERS' COMMENTS

Reviewer #1 (Remarks to the Author):

The manuscript by Freitas-Andrade et al. examines the developmental dynamics of gliovascular unit maturation with a specific focus on changes in astrocyte morphology and endfeet polarization around cortical blood vessels at the postnatal development. The authors aim to understand the cellular mechanisms underlying structural and transcriptional changes in mouse postnatal astrocytes in the brain, and how they relate to the timing of association of astrocyte endfeet with cerebral blood vessels. They identify that Hmgb1 is highly expressed by early postnatal astrocytes. Ablation of Hmgb1 in newborn astrocytes disrupts Connexin-43 distribution, alters astrocyte endfeet coverage of cerebral blood vessels as well as induces changes in endothelial cells at the ultrastructural level and neurovascular coupling into adulthood.

This is an elegant study that advances our understanding of how gliovascular unit maturation occurs in the developing brain. There is very little information in the literature about the formation of astrocyte endfeet (the ensheathment of blood vessels by astrocytes) and the mechanisms regulating this process. The authors provide new and interesting mechanical insights into cellular processes regulating astrocyte morphology during gliovascular unit maturation in the postnatal cortex by Hmgb1 through an extensive analyses of the Hmgb1 phenotype and potential downstream targets. Overall, the study will be of great interest to neuroscientists in particular neurovascular and glia biologists.

There are only a few minor points that the authors could address to strengthen some of the arguments for the paper:

1. The authors show that the total levels of Connexin-43 are reduced in the mutant (Supplementary Figure 10) and there is more Connexin-43+ in the astrocytes cell bodies in the P50 Hmgb1-DeltaAstro cortex compared to the wild-type mice. These data would suggest that the trafficking of Connexin-43 to the endfeet is affected in the Hmgb1-DeltaAstro astrocytes. The authors could address this in more depth in vitro by staining control and Hmgb1-DeltaAstro astrocytes for Connexin-43 and Aqp4 to determine their distribution in the cell as shown in other studies (Menezes et al., J Neurosci 2014; doi: 10.1523.JNEUROSCI.3678-13.2014.).

2. The authors provide compelling and clear data that cytoskeletal organization is affected in Hmgb1-DeltaAstro astrocytes, and these data are correlated with the downregulation of Ptger4, Slit1, Camk1g and Arhgef28 identified by bulk RNA sequencing. The authors could knockdown one of these genes (especially Ptger4) in astrocytes in vitro to show that it can mimick some of the phenotypes seen with the Hmgb1 KO astrocytes.

3. Hmgb1- DeltaAstro astrocytes have smaller Aldh1l1+ cell area compared to wild-type astrocytes (Figure 5a). However, Hmgb1-DeltaAstro astrocytes have larger GFAP+ cell area in vitro compared to wild-type astrocytes (Figure 5d). Does this reflect increased reactive astrocyte phenotype in vitro? Is there any indication of increased astrocyte reactivity from the transcriptome data in vitro?

Reviewer #3 (Remarks to the Author):

As already widely discussed, the results in Fig 1h have been published previously by Gilbert et al

2021.

Therefore, the correct sentence is "Overall, this shows that the perivascular placement of the endfoot establishes around P7 and is complete at P14" AS PREVIOUSLY SHOWED BY GILBERT ET AL.

All previous comments have been taken into account.

We would like to thank both reviewers for their constructive feedback, and we are glad to provide a final version addressing their concerns. Please find our detailed responses below (in blue). *Reviewers' comments are in black font italicized.*

Response to Reviewer #1:

The manuscript by Freitas-Andrade et al. examines the developmental dynamics of gliovascular unit maturation with a specific focus on changes in astrocyte morphology and endfeet polarization around cortical blood vessels at the postnatal development. The authors aim to understand the cellular mechanisms underlying structural and transcriptional changes in mouse postnatal astrocytes in the brain, and how they relate to the timing of association of astrocyte endfeet with cerebral blood vessels. They identify that Hmgb1 is highly expressed by early postnatal astrocytes. Ablation of Hmgb1 in newborn astrocytes disrupts Connexin-43 distribution, alters astrocyte endfeet coverage of cerebral blood vessels as well as induces changes in endothelial cells at the ultrastructural level and neurovascular coupling into adulthood.

This is an elegant study that advances our understanding of how gliovascular unit maturation occurs in the developing brain. There is very little information in the literature about the formation of astrocyte endfeet (the ensheathment of blood vessels by astrocytes) and the mechanisms regulating this process. The authors provide new and interesting mechanical insights into cellular processes regulating astrocyte morphology during gliovascular unit maturation in the postnatal cortex by Hmgb1 through an extensive analyses of the Hmgb1 phenotype and potential downstream targets. Overall, the study will be of great interest to neuroscientists in particular neurovascular and glia biologists.

There are only a few minor points that the authors could address to strengthen some of the arguments for the paper. We sincerely thank the first Reviewer for recognizing the impact of our study.

1. *The authors show that the total levels of Connexin-43 are reduced in the mutant (Supplementary Figure 10) and there is more Connexin-43+ in the astrocytes cell bodies in the P50 Hmgb1-DeltaAstro cortex compared to the wild-type mice. These data would suggest that the trafficking of Connexin-43 to the endfeet is affected in the Hmgb1-DeltaAstro astrocytes. The authors could address this in more depth in vitro by staining control and Hmgb1-DeltaAstro astrocytes for Connexin-43 and Aqp4 to determine their distribution in the cell as shown in other studies (Menezes et al., J Neurosci 2014; doi: 10.1523/JNEUROSCI.3678-13.2014.). Indeed, there were not more Cx43+ astrocytes cell bodies at P50. What the graph shows (in Figure 7b), is that the size (i.e. area) of the "Cx43-high" patches significantly increase between P14 and P50. It means that, at P50, the mutant astrocytes still present this interesting dense punctate Cx43 staining. However, since this is a subset of astrocytes, we believe that this does not influence the overall protein levels of Cx43 in the cerebral cortex detected by WB at P14. We believe that Cx43 localization/distribution is disrupted in the astrocytes, with varying degrees. This "uncoupling" between IHC and WB data suggests that the distribution/targeting of CX43 to endfeet is affected, much probably because of changes in the cytoskeleton that we find. We propose that the actin rings prevent the acquisition of a normal astrocyte morphology, and subsequently prevent the normal distribution of proteins along extensions towards endfeet. We agree with the Reviewer that staining for Cx43 and AQP4 in culture (ICC) would further support this interpretation, however this would imply setting up new breedings, waiting to obtain enough control and mutant littermates, extracting more primary astrocytes, altogether requiring considerable time and resources. We will consider these excellent ideas for follow-up work.*

2. The authors provide compelling and clear data that cytoskeletal organization is affected in *Hmgb1-DeltaAstro* astrocytes, and these data are correlated with the downregulation of *Ptger4*, *Slit1*, *Camk1g* and *Arhgef28* identified by bulk RNA sequencing. The authors could knockdown one of these genes (especially *Ptger4*) in astrocytes *in vitro* to show that it can mimick some of the phenotypes seen with the *Hmgb1 KO* astrocytes. Point very well taken, we thank the Reviewer for this excellent suggestion. While we agree that this is an important experiment to perform, we believe it goes beyond the scope of the current study (in which we reveal a new molecular player in gliovascular maturation). We will keep this idea for a follow-up grant application.

3. *Hmgb1-DeltaAstro* astrocytes have smaller *Aldh1l1+* cell area compared to wild-type astrocytes (Figure 5a). However, *Hmgb1-DeltaAstro* astrocytes have larger GFAP+ cell area *in vitro* compared to wild-type astrocytes (Figure 5d). Does this reflect increased reactive astrocyte phenotype *in vitro*? Is there any indication of increased astrocyte reactivity from the transcriptome data *in vitro*? In fact, the use of GFAP to visualize astrocytes *in vitro* is well accepted, as primary mouse brain astrocytes grown in culture produce more GFAP, which is detectable by ICC (e.g., PMID: 23380713, 31989426, 34917973). This is much probably due to their reactivity to culture serum conditions. And of note, we show that mutant astrocytes are not reactive *in vivo*, as illustrated by low GFAP immunoreactivity in both mutant and control mice (see Supplementary Figure 10b). Moreover, our primary mouse brain astrocytes used for cell cultures or RNA sequencing display comparable expression levels of *Gfap* (Supplementary Fig. 9b). We have attempted to stain astrocytes in culture using *Aldh1L1* antibodies, but while it worked very well in tissue sections we were unsuccessful at producing a convincing staining in culture conditions.

While the Reviewer raises a good point regarding the discrepancy in cell area between *in vivo* and *in vitro* conditions, we believe that this difference is due to the different environment in which astrocytes are found. *In vitro*, astrocytes are not restrained/compacted by surrounding cells (neurons, microglia, oligodendrocytes, etc). Hence, with cytoskeletal dysregulation, they can adopt a more flattened shape *in vitro*. Moreover, consistent with our *in vivo* data, mutant astrocytes in culture display a less complex morphology with far less processes compared to control astrocytes, supporting our interpretation.

Response to Reviewer #3:

As already widely discussed, the results in Fig 1h have been published previously by Gilbert et al 2021.

Therefore, the correct sentence is "Overall, this shows that the perivascular placement of the endfoot establishes around P7 and is complete at P14" AS PREVIOUSLY SHOWED BY GILBERT ET AL.

In order to properly acknowledge this work, we have corrected our sentence to: "*Altogether this shows that perivascular endfoot placement establishes around P7 and is complete by P14, in line with observations made by Gilbert et al (2021).*"

All previous comments have been taken into account. Thank you.